# SECOND-MOMENT LOSS: A NOVEL REGRESSION OBJECTIVE FOR IMPROVED UNCERTAINTIES

## ABSTRACT

Quantification of uncertainty is one of the most promising approaches to establish *safe* machine learning. Despite its importance, it is far from being generally solved, especially for neural networks. One of the most commonly used approaches so far is Monte Carlo dropout, which is computationally cheap and easy to apply in practice. However, it can underestimate the uncertainty. We propose a new objective, referred to as second-moment loss (SML), to address this issue. While the full network is encouraged to model the mean, the dropout networks are explicitly used to optimize the model variance. We analyze the performance of the new objective on various toy and UCI regression datasets. Comparing to the state-of-the-art of deep ensembles, SML leads to comparable prediction accuracies and uncertainty estimates while only requiring a single model. Under distribution shift, we observe moderate improvements. From a safety perspective also the study of worst-case uncertainties is crucial. In this regard we improve considerably. Finally, we show that SML can be successfully applied to SqueezeDet, a modern object detection network. We improve on its uncertainty-related scores while not deteriorating regression quality. As a side result, we introduce an intuitive Wasserstein distance-based uncertainty measure that is non-saturating and thus allows to resolve quality differences between any two uncertainty estimates.

## 1 INTRODUCTION

Having attracted great attention in both academia and digital economy, deep neural networks (DNNs, Goodfellow et al. (2016)) are about to become vital components of safety-critical applications. Examples are autonomous driving (Pomerleau, 1989; Bojarski et al., 2016) or medical diagnostics (Liu et al., 2014), where prediction errors potentially put humans at risk. These systems require methods that are robust not only under lab conditions (i.i.d. data sampling), but also under continuous domain shifts, think e.g. of adults on e-scooters or growing numbers of mobile health sensors. Besides shifts in the data, the data distribution itself poses further challenges. Critical situations are (fortunately) rare and thus strongly under-represented in datasets. Despite their rareness, these critical situations have a significant impact on the safety of operations. This calls for comprehensive self-assessment capabilities of DNNs and recent uncertainty mechanisms can be seen as a step in that direction.

While a variety of uncertainty approaches has been established, stable quantification of uncertainty is still an open problem. Many recent machine learning applications are e.g. equipped with Monte Carlo (MC) dropout (Gal & Ghahramani, 2016) that offers conceptual simplicity and scalability. However, is tends to underestimate uncertainties thus bearing disadvantages compared to more recent approaches such as deep ensembles (Lakshminarayanan et al., 2017). We propose an alternative uncertainty mechanism. It builds on dropout sub-networks and explicitly optimizes variances (see Fig. 1 for an illustrative example). Technically, this is realized by a simple additive loss term, the *second-moment loss*. To address the above outlined requirements for safety-critical systems, we evaluate our approach systematically w.r.t. continuous data shifts and worst-case performances.

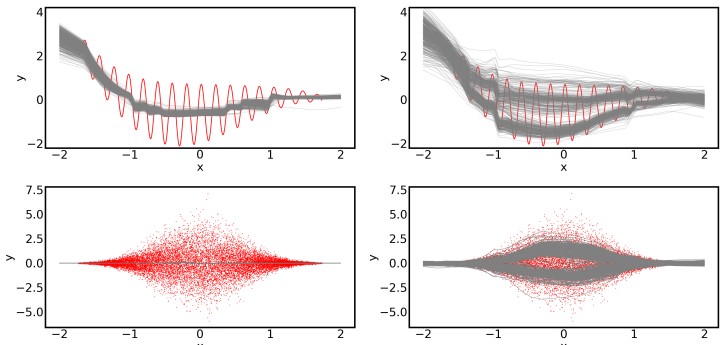

Figure 1: Sampling-based uncertainty mechanisms on toy datasets. In contrast to MC dropout (left), the second-moment loss (right) induces uncertainties that capture aleatoric uncertainty. The ground truth data is shown in red. Each grey line represents the outputs of one of 200 random sub-networks that are obtained by applying dropout-based sampling to the trained full network. For details on the data sets ('toy-hf', 'toy-noise'), the neural architecture and the uncertainty methods please refer to section 4 and references therein.

In detail, our contribution is as follows:

- we introduce a novel regression loss for better calibrated uncertainties applicable to dropout networks,

- we reach state-of-the-art performance in an empirical study and improve on it when considering data shift and worst-case performances, and

- we demonstrate its applicability to real-world applications by example of 2D bounding box regression.

## 2 RELATED WORK

Approaches to estimate predictive uncertainties can be broadly categorized into three groups: Bayesian approximations, ensemble approaches and parametric models.

Monte Carlo dropout (Gal & Ghahramani, 2016) is a prominent representative of the first group. It offers a Bayesian motivation, conceptual simplicity and scalability to application-size neural networks (NNs). This combination distinguishes MC dropout from other Bayesian neural network (BNN) approximations like Blundell et al. (2015) and Ritter et al. (2018). A computationally more efficient version of MC dropout is one-layer or last-layer dropout (see e.g. Kendall & Gal (2017)). Alternatively, analytical moment propagation allows sampling-free MC-dropout inference at the price of additional approximations (e.g. Postels et al. (2019)). Further extensions of MC dropout target tuned performance by learning layer-specific drop rates using Concrete distributions (Gal et al., 2017) and the integration of aleatoric uncertainty (Kendall & Gal, 2017). Note that dropout training is used—independent from an uncertainty context—for better model generalization (Srivastava et al., 2014).

Ensembles of neural networks, so-called deep ensembles (Lakshminarayanan et al., 2017), pose another popular approach to uncertainty modelling. Comparative studies of uncertainty mechanisms (Snoek et al., 2019; Gustafsson et al., 2020) highlight their advantageous uncertainty quality, making deep ensembles a state-of-the-art method. Fort et al. (2019) argue that deep ensembles capture multi-modality of loss landscapes thus yielding potentially more diverse sets of solutions.

The third group are parametric modelling approaches that extend point estimations by adding a model output that is interpreted as variance or covariance (Nix & Weigend, 1994; Heskes, 1997). Typically, these approaches optimize a (Gaussian) negative log-likelihood (NLL, Nix & Weigend (1994)). A more recent representative of this group is, e.g., Kendall & Gal (2017), for a review see Khosravi et al. (2011). A closely related model class is deep kernel learning. It approaches uncertainty modelling by combining NNs and Gaussian processes (GPs) in various ways, e.g. via

an additional layer (Wilson et al., 2016; Iwata & Ghahramani, 2017), by using networks as GP kernels (Garnelo et al., 2018) or by matching NN residuals with a GP (Qiu et al., 2019).

In the context of object detection, various uncertainty approaches can be encountered, e.g. MC dropout in Bhattacharyya et al. (2018) and Miller et al. (2018), or parametric approaches in He et al. (2019). Hall et al. (2020) advocate to account for uncertainty in bounding box detection.

The quality of uncertainties is typically evaluated using negative log-likelihood (Blei et al., 2006; Walker et al., 2016; Gal & Ghahramani, 2016), expected calibration error (ECE) (Naeini et al., 2015; Snoek et al., 2019) and its variants and by considering correlations between uncertainty estimates and model errors, e.g. area under the sparsification error curve (AUSE, Ilg et al. (2018)) for image tasks. Moreover, it is common to study how useful uncertainty estimates are for solving auxiliary tasks like out-of-distribution classification (Lakshminarayanan et al., 2017) or robustness w.r.t. adversarial attacks. An alternative approach is the investigation of qualitative uncertainty behaviors: Kendall & Gal (2017) check if the epistemic uncertainty decreases when increasing the training set or Wirges et al. (2019) studies how the level of uncertainty depends on the distance of the object to a car for some 3D environment regression task.

## 3 SECOND-MOMENT LOSS

Monte Carlo (MC) dropout was proposed as a computationally cheap approximation of performing Bayesian inference in neural networks (Gal & Ghahramani, 2016). Given a neural network $f_\theta : \mathbb{R}^d \to \mathbb{R}^m$ with parameters $\theta$, MC dropout samples sub-networks $f_{\tilde{\theta}}$ by randomly dropping nodes from the main model $f_\theta$. During MC dropout inference the prediction is given by the mean estimate over the predictions of a given sample of sub-networks, while the uncertainty associated with this prediction can be estimated, e.g. , in terms of the sample variance. During MC dropout training the objective function, e.g. , (in our case) the mean squared error (MSE), is applied to the sub-networks separately. Due to this training procedure, all sub-network predictions are shifted towards the same training targets, which can result in overconfident predictions, i.e. in an underestimation of prediction uncertainty.[1]

Based on this observation, we propose to use the sub-networks $f_{\tilde{\theta}}$ in a different way: they are explicitly *not* encouraged to fit the data mean directly. This is the task of the full network $f_\theta$. The sub-networks $f_{\tilde{\theta}}$ instead model aleatoric uncertainty and prediction residuals if the prediction of the full network $f_\theta$ is incorrect. Thus, we deliberately assign different 'jobs' to the main network $f_\theta$ on the one hand and its sub-networks on the other hand. Formalizing this idea into an optimization objective yields

$$ L = L_{\text{regr}} + L_{\text{sml}} = \frac{1}{M} \sum_{i=1}^{M} \Big[ \underbrace{(f_\theta(x_i) - y_i)^2}_{\text{regression loss}} + \beta \underbrace{(|f_{\tilde{\theta}}(x_i) - f_\theta(x_i)| - |f_\theta(x_i) - y_i|)^2}_{\text{second-moment loss}} \Big] \ , \quad (1) $$

where the sum runs over a mini-batch of size $M < N$ taken from the set of observed samples $\mathcal{D} = \{(x_i, y_i)\}_{i=1}^N$, $x_i \in \mathbb{R}^d$ denotes the input, $y_i \in \mathbb{R}^m$ the ground-truth label, and $\beta > 0$ is a hyper-parameter that weights both terms. The first term, $L_{\text{regr}}$, is the MSE w.r.t. the full network $f_\theta$. The second term, $L_{\text{sml}}$, seeks to optimize[2] the sub-networks $f_{\tilde{\theta}}$. It aims at finding sub-networks such that the distance $|f_{\tilde{\theta}}(x_i) - f_\theta(x_i)|$ matches the prediction residual, quantified by $|f_\theta(x_i) - y_i|$, which also serves as a proxy to the aleatoric uncertainty (compare Fig. 2, top). As our choice of $L_{\text{sml}}$ removes all directional information of the residual, possible (optimal) solutions for the $f_{\tilde{\theta}}$ are not uniquely determined.[3] This leads to a significant increase in the variance of the sub-networks, i.e. the second moment of $f_{\tilde{\theta}}$, compared to standard MC dropout, which is why we name $L_{\text{sml}}$ the *second-moment loss* (SML).[4] The standard deviations $\sigma_{\text{total}}$ of the predictions of the sub-networks w.r.t.

---

[1] An intuitive explanation is as follows: Let $f_\theta$ be a NN with one-dimensional output. For MC dropout with the MSE loss we get $\langle (f_{\tilde{\theta}}(x) - y)^2 \rangle = (\langle f_{\tilde{\theta}}(x) \rangle - y)^2 + \sigma^2(f_{\tilde{\theta}}(x))$. Therefore, it simultaneously minimizes the squared error between sub-network mean and target and the variance $\sigma^2(f_{\tilde{\theta}}(x)) = \langle f_{\tilde{\theta}}^2(x) \rangle - \langle f_{\tilde{\theta}}(x) \rangle^2$ over the sub-networks.

[2] To avoid unintended optimization of full $f_\theta$ in direction of $f_{\tilde{\theta}}$, we only back-propagate through $f_{\tilde{\theta}}$ in $L_{\text{sml}}$.

[3] For an analytical study of the loss landscape induced by $L_{\text{sml}}$ see appendix A.1.

[4] For brevity, we also refer to the entire loss objective $L$ as second-moment loss during evaluation.

the prediction of the mean network induced by the SML have two components: the spread $\sigma_{\mathrm{drop}}$ of the sub-networks and an offset $\left|f_\theta - \langle f_{\tilde{\theta}} \rangle\right|$ between the full network and the sub-network mean that our loss might cause, concretely, $\sigma_{\mathrm{total}} = \sigma_{\mathrm{drop}} + |f_\theta - \langle f_{\tilde{\theta}} \rangle|$. While $|f_\theta - \langle f_{\tilde{\theta}} \rangle|$ is reminiscent of residual matching, $\sigma_{\mathrm{drop}}$ seems to be more closely related to modelling uncertainties. We show in appendix A.2 that $\sigma_{\mathrm{drop}}$ accounts on average for more than $80\%$ of $\sigma_{\mathrm{total}}$ in our experiments.

Note that while we investigate the proposed objective in terms of dropout sub-networks in this paper, our arguments as well as the actual approach are generally applicable to other models that allow to formulate sub-networks given some kind of mean model. Besides the regression tasks considered here our approach could be useful for other objectives which use or benefit from an underlying distribution, e.g. uncertainty quantification in classification.

## 4 Experiments

We begin this section with an illustrative and visualizable toy dataset (section 4.1) and continue with benchmarks on various UCI datasets (Dua & Graff, 2017) in section 4.2. To conclude in 4.3, the second-moment loss is applied to a more complex task: object detection in the form of a 2D bounding box regression using the compact SqueezeDet architecture (Wu et al., 2017). Technical details for this section are relegated to appendix B.

For the first two parts we use an identical set-up of 2 hidden layers with 50 neurons each and ReLu activations. As benchmark methods we consider: MC dropout (abbreviated as **MC**), last-layer MC dropout (**MC-LL**), parametric uncertainty (**PU**), deep ensembles with (**PU-DE**) and without (**DE**) explicit PU and PU combined with MC (**PU-MC**). All considered types of networks provide estimates $(\mu_i, \sigma_i)$, where $\sigma_i$ is obtained directly as model output (PU), by sampling (MC, MC-LL, SML) or as an ensemble aggregate (DE, PU-DE). For PU-MC, a combination of parametric output and sampling is employed.

While the toy model has a stronger focus on visual inspection, the UCI evaluation relies on a variety of measures: root-mean-square error (**RMSE**), negative log-likelihood (**NLL**), expected calibration error (**ECE**), and a novel usage of the Wasserstein distance (**WS**). In using a least squares regression, we make the standard assumption that errors follow a Gaussian distribution. This assumption is reflected in the (standard) definitions of above named measures, i.e. all uncertainty measures quantify the set of outputs $\{(\mu_i, \sigma_i)\}$ relative to a Gaussian distribution.[5] Details on the definition of those measures, as well as on the network and training procedure and the implementation of methods can be found in appendix B.1.

### 4.1 Toy datasets

To illustrate qualitative behaviors of the different uncertainty techniques, we consider two $\mathbb{R} \to \mathbb{R}$ toy datasets. This benchmark puts a special focus on the handling of data-inherent uncertainty. The first dataset is Gaussian white noise with an $x$-dependent (non-linear) amplitude, see first row of Fig. 2. The second dataset is a polynomial overlayed with a high-frequency, amplitude-modulated sine, see fourth row of Fig. 2. The explicit equations for the toy datasets used here can be found in appendix B.2. While the uncertainty in the first dataset ('toy-noise') is clearly visible, it is less obvious for the fully deterministic second dataset ('toy-hf'). There is an effective uncertainty though, as the shallow networks employed are empirically not able to fit (all) fluctuations of 'toy-hf' (see fifth row of Fig. 2). One might (rightfully) argue that this is a sign of insufficient model capacity. But, in more realistic, e.g., higher dimensional and sparser datasets the distinction between true noise and complex information becomes exceedingly difficult to make. As the Nyquist-Shannon sampling theorem states, with limited data deterministic fluctuations above a cut-off frequency can no longer be resolved (Landau, 1967). They therefore become virtually indistinguishable from random noise.

The mean estimates of all uncertainty methods (second and fifth row in Fig. 2) look alike on both datasets. They approximate the noise mean and the polynomial, respectively. In the latter case, all methods rudimentarily fit some individual fluctuations. The variance estimation (third and sixth row in Fig. 2) in contrast reveals significant differences between the methods: While PU, PU-

---

[5]While different distributions, e.g. exponentially decaying or mixtures, could be used, we restrict the scope here to the standard Gaussian case.

DE, and the network trained with SML are capable of capturing aleatoric uncertainty, MC dropout variants and non-parametric ensembles are not. This behavior of MC dropout is expectable as it was introduced to account for model uncertainty not data-inherent uncertainty. The non-parametric ensemble is effectively optimized in a similar fashion. In contrast, NLL-optimized PU networks have a home-turf advantage on these datasets since the parametric variance is explicitly optimized to account for the present aleatoric uncertainty. The SML provides comparably good uncertainty estimates. They are evoked by the $L_{\text{sml}}$-term that incentivizes sub-networks $f_{\tilde{\theta}}$ to keep an adequate distance from $f_{\theta}$. The results on the 'toy-hf' dataset exemplify that the SML can provide good uncertainty estimates even for networks with insufficient expressiveness. While the outcomes of both PU (PU-DE) and SML-trained network look similar, the mechanics of the two approaches are fundamentally different. We investigate the drivers behind the adjustments of the sub-networks in appendices A and A.3. Accompanying quantitative evaluations can be found in appendix B.2.

In the following, we substantiate the corroborative results of the SML on toy data by an empirical study on UCI datasets and an application to a modern object detection network.

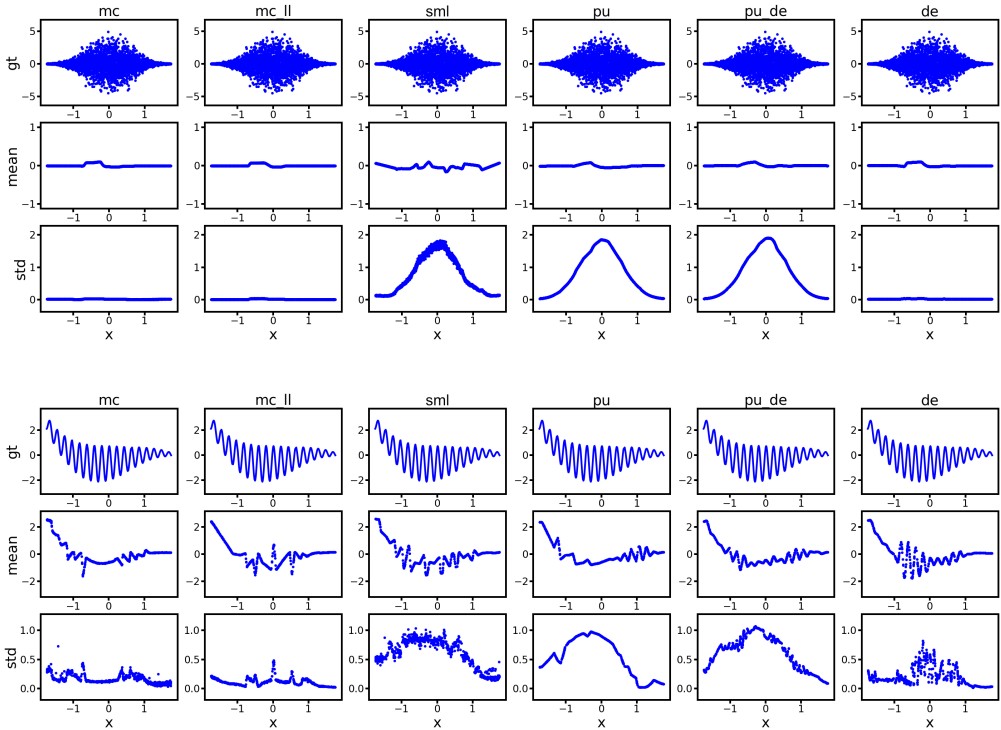

Figure 2: Comparison of uncertainty approaches (columns) on two 1D toy functions: a noisy one (top) and a high-frequency one (bottom). Test data ground truth (respective first row) is shown with mean estimates (resp. second row) and standard deviations (resp. third row).

## 4.2 UCI REGRESSION DATASETS

Next, we study UCI regression datasets, extending the dataset selection in Gal & Ghahramani (2016) by adding three further datasets: 'diabetes', 'california', and 'superconduct'. Apart from train- and test-data results, we study regression performance and uncertainty quality *under data shift*. Such distributional changes and uncertainty quantification are closely linked since the latter ones are rudimentary "self-assessment" mechanisms that help to judge model reliability. These judgements gain importance for model inputs that are *structurally different* from train data. Appendix B.3 elaborates on different ways of splitting the data, namely *pca-based* splits in input space (using the first principal component) and *label-based* splits. For both we consider *interpolation testing* and *extrapolation testing*, where the training omits data within or at the edges of the full data range, respectively. For example, for labels running from 0 to 1, (label-based) extrapolation testing would consider only data

with a label larger 0.9, while training would be performed on the smaller label values. More general information on training and dataset-dependent modifications to the experimental setup are relegated to the technical appendix B.1. For brevity of exposition, we limit our discussion here largely to the ECE and the worst-case uncertainty performance. An evaluation of the remaining measures, including the Wasserstein measure, is given in appendix B.3. All presented results are 5- or 10-fold cross validated.

Fig. 3 provides average ECE values of the outlined uncertainty methods under i.i.d. conditions (first and second panel), under label-based data shifts (third and fourth panel) and under pca-based data shifts (fifth and sixth panel). For the shifts we either use interpolation or extrapolation where we The visualized mean (median) values and quantile intervals are obtained by averaging over ECE values on 13 UCI datasets. On training data, ECEs are smallest for PU, followed by PU-MC, PU-DE and the SML-trained network. On test data, however, PU-MC, PU-DE and the SML-trained network share the first place. Looking at the stability w.r.t. data shift, i.e. extra- and interpolation based on label-split or pca-split, PU loses in performance while PU-DE, PU-MC and SML reach the smallest calibration errors. Regarding the 75% quantiles, SML consistently provides one of the best result on the standard test set and on all out-of-data test sets.

For NLL and Wasserstein measure, PU-DE, PU-MC and the SML-trained network reach comparably small average values with advantages for the SML-trained network under data shift, see Fig. 9 and Fig. 10 in appendix B.3 for detailed evaluations. Especially for NLL a strong difference can be seen with respect to the consistency of performance under label-based data shifts, which suggests that SML is more "reliable" compared to PU based approaches. In contrast to uncertainty quality, regression performances are almost identical for all uncertainty methods (see Fig. 8 and Table 5 in appendix B.3).

Summarizing these evaluations on UCI datasets, we find SML to be as strong as the state-of-the-art methods of PU-DE and PU-MC. In comparison to PU-DE, PU-MC and SML use only a single network compared to an ensemble of 5 networks. We moreover observe advantages for SML under PCA- and label-based data shifts. Three datasets lead to overestimated uncertainties for the SML, see discussion in appendix B.3. A visual tool to further inspect uncertainty quality are residual-uncertainty scatter plots as shown in appendix B.4. For a reflection on NLL and comparisons of the different uncertainty measures on UCI data see again appendix B.3.

From a safety perspective the study of worst-case uncertainties is crucial. A better understanding of these least appropriate uncertainties might allow to determine lower bounds on operation quality of safety-critical systems. For this we define *normalized residuals* $r_i = (\mu_i - y_i)/\sigma_i$ based on the prediction estimates $(\mu_i, \sigma_i)$ for a given data point $(x_i, y_i)$. We restrict our analysis to uncertainty estimates that *under-estimate* prediction residuals, i.e. $|r_i| \gg 1$. These cases might be more harmful than overly large uncertainties, $|r_i| \ll 1$, that likely trigger a conservative system behavior. We quantify worst-case uncertainty performance as follows: for a given (test) dataset, the absolute normalized residuals $\{|r_i|\}_i$ are calculated. We determine the $99\%$ quantile $q_{0.99}$ of this set and calculate the mean value over all $|r_i| > q_{0.99}$, the so-called expected tail loss at quantile $99\%$ (**ETL$_{0.99}$**) (Rockafellar & Uryasev, 2002). The ETL$_{0.99}$ measures the average performance of the worst performing $1\%$.

For both toy datasets and 12 UCI datasets, the test data ETL$_{0.99}$'s of all trained network are calculated, yielding a total of 105 ETL$_{0.99}$ values per uncertainty method. Table 1 reports the mean value and the maximal value of these ETL$_{0.99}$'s for PU-MC, PU-DE and SML-trained networks as these three methods show the strongest performances throughout this work. While none of these methods gets close to the ideal ETL$_{0.99}$'s of a $\mathcal{N}(0, 1)$, SML-trained networks exhibit significantly less pronounced tails and therefore higher stability compared to PU-MC and PU-DE. This holds true over all considered test sets. Deviations from standard normal grow from i.i.d. test over PCA-based train-test split to label-based train-test split. We attribute the lower stability of PU-DE to the nature of the PU networks composing the ensemble. The inherent instability of parametric uncertainty estimation (see Table 5 in appendix B.3) is largely suppressed by ensembling. Considering the tail of the $|r_i|$-distribution however reveals that regularization of PU by ensembling works not in every single case. It is unlikely that larger ensemble are able to fully cure this instability issue. Regularizing PU by applying dropout (PU-MC) leads to comparably weak results. SML-trained networks in contrast encode uncertainty into the structure of the entire network thus yielding preferable stability compared to parametric approaches.

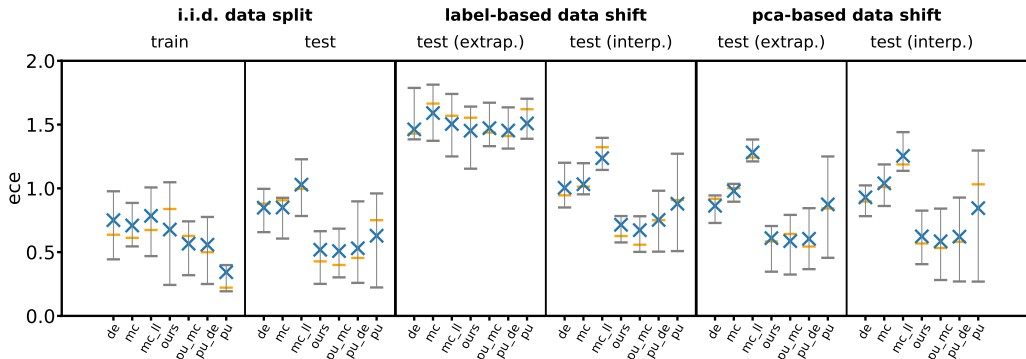

Figure 3: Expected calibration errors (ECEs) of different uncertainty methods under i.i.d. conditions (first and second panel) and under various kinds of data shift (third to sixth panel, see text for details). SML ('ours') is compared to 5 benchmark approaches. Each blue cross is the mean over ECE values from 13 UCI regression datasets. Orange line markers indicate median values. The gray vertical bars reach from the 25% quantile (bottom horizontal line) to the 75% quantile (top horizontal line).

Table 1: Worst-case uncertainty quality for different uncertainty methods: SML-induced uncertainties ('Ours'), PU-DE and PU-MC are compared to the ideal Gaussian case for i.i.d. and non-i.i.d. data splits. Worst-case uncertainty quality is quantified by the expected tail loss at the 99% quantile ($ETL_{0.99}$). Each mean and max value is taken over the ETLs of 105 models trained on 14 different datasets.

| measure | data split | $\mathcal{N}(0,1)$ | Ours | PU-DE | PU-MC |
|---|---|---|---|---|---|
| mean $ETL_{0.99}$ | i.i.d. | 2.89 | 3.80 | 5.03 | 5.55 |
| max $ETL_{0.99}$ | i.i.d. | 3.01 | 9.71 | 19.69 | 30.35 |
| mean $ETL_{0.99}$ | pca | 2.89 | 4.62 | 6.71 | 6.12 |
| max $ETL_{0.99}$ | pca | 3.01 | 13.0 | 39.34 | 21.58 |
| mean $ETL_{0.99}$ | label | 2.89 | 5.18 | 38.65 | 49.82 |
| max $ETL_{0.99}$ | label | 3.01 | 35.96 | 799.78 | 631.84 |

## 4.3 APPLICATION TO OBJECT REGRESSION

After studying toy and UCI datasets, we turn towards the challenging real-world task of object detection, namely the SqueezeDet model (Wu et al., 2017), a fully convolutional neural network. It is trained and evaluated on KITTI (Geiger et al., 2012). For details on the SqueezeDet architecture and the KITTI data split, see B.5. We compare standard SqueezeDet with SML-SqueezeDet that uses the second-moment loss instead of the original MSE regression loss (see appendix B.5 for more details). In both settings the model is trained for $150,000$ mini-batches of size 20, i.e. for 815 epochs. After training, we keep dropout active and compute 50 forward passes for each test image. For standard SqueezeDet, all forward passes are individually matched with ground truth. We exclude predictions from the evaluation if their IoU with ground truth is $\leq 0.1$. While standard SqueezeDet (with activated dropout at inference) uses the mean of the dropout samples for prediction, SML-SqueezeDet uses the full network instead (see section 3). These predictions and their corresponding dropout samples are matched based on the respective anchor. The dropout samples are summarized by their means and variances.

To assess model performance, we report the mean intersection over union (mIoU) and RMSE (in pixel space) between predicted bounding boxes and matched ground truths. The quality of the uncertainty estimates is measured by (coordinate-wise) NLL, ECE and Wasserstein distance. Table 2 shows a summary of our results on train and test data. The results for NLL, ECE and WS have been

averaged across the 4 regression coordinates. SqueezeDet and SML-SqueezeDet show comparable regression results, with slight advantages for SML-SqueezeDet on test data. Considering uncertainties quality, we find substantial advantages for SML-SqueezeDet across all evaluation measures. These findings resemble those on the UCI regression datasets and indicate that the second-moment loss works well on a modern application-scale network.

Table 2: Regression performance and uncertainty quality of SqueezeDet-type networks on KITTI test data. SML-trained SqueezeDet (ours) is compared with the default SqueezeDet that uses one-layer dropout to estimate uncertainties. The measures of NLL, ECE and WS are aggregated along their respective four dimensions, for details see appendix B.5 and Table 6 therein.

| measure | SqueezeDet | SML-SqueezeDet | SqueezeDet | SML-SqueezeDet |
|---|---|---|---|---|
| | train | | test | |
| mIoU ($\uparrow$) | 0.816 | 0.812 | 0.738 | 0.744 |
| RMSE ($\downarrow$) | 6.418 | 6.862 | 18.225 | 17.492 |
| NLL ($\downarrow$) | 20.746 | 3.916 | 98.807 | 17.875 |
| ECE ($\downarrow$) | 0.996 | 0.554 | 1.198 | 0.834 |
| WS ($\downarrow$) | 2.487 | 0.874 | 4.587 | 1.734 |

## 5 CONCLUSION

We approach dropout-based uncertainty quantification from a new direction: sub-networks are explicitly not encouraged to model the data mean, they capture data-inherent uncertainties and potential fitting residuals of the full network instead. Technically, this is realized by an additional loss term that accompanies the standard regression objective: the *second-moment loss*. Our loss enables stable training. Training complexity and runtime behavior at inference are comparable to MC dropout. Task performances and uncertainty qualities of these models are on par with (parametric) deep ensembles, the widely used state-of-the-art for uncertainty quantification. However, unlike deep ensembles, we use single networks. In practice, this might allow to reduce training effort significantly compared to deep ensembles, especially for application-scale networks. Moreover, a single network requires only a fraction of the storage of a deep ensemble, making models with competitive uncertainties more accessible for mobile or embedded applications.

An extensive study of uncertainties under data shift revealed advantages of SML-trained models compared to deep ensembles: while both methods *on average* provide comparable results, we find a higher stability across a variety of datasets and data shifts. With respect to worst-case uncertainties SML-trained networks are by a large margin better than deep ensembles. A quite relevant finding for safety-critical applications like automated driving or medical diagnosis where (even rarely occurring) inadequate uncertainty estimates might lead to injuries and damage. Technically, we attribute this gain in stability to our sub-network-based approach: like MC dropout, we integrate uncertainty estimates into the very structure of the network, rendering it more robust towards unseen inputs than a parameter estimate.

Moreover, the second-moment loss can serve as a general drop-in replacement for MC dropout on regression tasks. For already trained MC dropout models, post-training with the second-moment loss might suffice to improve on uncertainty quality. As an outlook, our first such post-training experiments on UCI datasets are encouraging. Another interesting variant is the combination of SML with last-layer dropout as it enables sampling-free inference (Postels et al., 2019). Preliminary experiments on UCI datasets show clearly improved uncertainties qualities compared to standard MC-LL. A potentially interesting avenue for near-real time applications.

The simple additive structure of the second-moment loss makes it applicable to a variety of optimization objectives. For classification, we might be able to construct a non-parametric counterpart to prior networks (Malinin & Gales, 2018). Taking a step back, we demonstrated an easily feasible approach to influence and train sub-network distributions. This could be a promising avenue, for distribution matching but also for theoretical investigations.

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

# SUPPLEMENTARY MATERIAL

This part accompanies our paper "*Second-Moment Loss: A Novel Regression Objective for Improved Uncertainties*" and provides further in-depth information. In Section A we provide both theoretical and numerical insight into the resulting uncertainties of our loss modification. Large parts of the empirical evaluation on UCI can be found in section B, including details on the setup, data splits as well as further uncertainty measures. Details for our use and modification of SqueezeDet are located in sub-section B.5. As the second-moment loss couples to the usual MSE regression loss via a hyper-parameter $\beta$ we test various values in section C, finding no strong correlation between result and parameter. We close with a discussion on the relation between uncertainty measures and their respective sensitivity in section D.

## A   MECHANICS OF THE SECOND-MOMENT LOSS

We analytically study the optimization landscape evoked by the second-moment loss in A.1. This analysis provides building blocks to better understand the composition of the SML-uncertainties as detailed in the remainder of this section.

### A.1   ANALYTICAL PROPERTIES OF THE SECOND-MOMENT LOSS

In the following, we look closer at the behaviour of the second-moment loss with respect to aleatoric uncertainty. For this, we assume that the residuals, compare eq. (1), are given by a Gaussian distribution with, for simplicity, $\mu_{\text{Res.}} = 0$ and $\sigma_{\text{Res.}} = 1$. We want to determine the resulting loss for the $L_{\text{sml}}$ term in eq. (1) governing the uncertainty estimation of the model. It depends on the underling distribution of the effective MC dropout distribution, which me model as $\mathcal{N}(\mu_{\text{drop}}, \sigma_{\text{drop}})$ such that:

$$L_{\text{sml}} = \int_{-\infty}^{\infty} \mathrm{d}y_1 \mathrm{d}y_2 \left(|y_1| - |y_2|\right)^2 p_1(y_1) \, p_2(y_2)\,, \tag{2}$$

where $p_1$ and $p_2$ are the Gaussian distributions discussed above. After some calculation this yields:

$$L_{\text{sml}} = -\frac{4}{\pi}\sigma_{\text{drop}} \exp\left(-\frac{1}{2}\frac{\mu_{\text{drop}}^2}{\sigma_{\text{drop}}^2}\right) - \sqrt{\frac{8}{\pi}}\,\mu_{\text{drop}}\,\text{Erf}\left(\frac{\mu_{\text{drop}}}{\sqrt{2}\,\sigma_{\text{drop}}}\right) + \sigma_{\text{drop}}^2 + \mu_{\text{drop}}^2 + 1\,, \tag{3}$$

which is visualized in Fig. 4. The two global minima can be found for $\sigma_{\text{drop}} = 0$ and $\mu_{\text{drop}} = \pm\sqrt{2/\pi}$. However, as we model a randomized residual $y_1$ these minima do not reach zero. We find that it is favourable to move $\mu_{\text{drop}}$ away from the network prediction of $\mu_{\text{Res.}} = 0$, the mean of the underlying data distribution. But, this is only the case as long as the inherent uncertainty in the dropout distribution can be brought below $\sigma_{\text{drop}} < 2/\pi$, which is still smaller than the uncertainty of $\sigma_{\text{Res.}} = 1$ assumed within the training data distribution. Otherwise, it is more favourable to have $\mu_{\text{drop}} = \mu_{\text{Res.}} = 0$. In the following sections we investigate the practical implications of this finding. For instance, as detailed in section B.2, the highly oscillating or noisy toy model experiments clearly exhibited the type of separation discussed here. Decomposing the uncertainty for the UCI datasets in section A.2, on the other hand, showed mixed behaviour with indications for bi-modal shifts in $\mu_{\text{drop}}$ as well as improved values of $\sigma_{\text{drop}}$.

We already showed the effect of this bi-modality in Fig. 1 at the beginning of the paper, where various sub-networks where sampled. Clearly visible is a stronger variation between the networks compared to MC, but also a concentration around the two possible minima. While this Fig. provides a good visual estimate of $\sigma_{\text{drop}}$ the total uncertainty $\sigma_{\text{total}}$ would additionally contain the systematic shift $|f_\theta - \langle f_{\tilde\theta}\rangle|$. Given the roughly symmetric distribution of the sub-networks we can expect it to be comparatively small.

### A.2   COMPOSITION OF THE UNCERTAINTY ESTIMATE

The uncertainty estimate of the second-moment loss is comprised of two parts: $\sigma_{\text{total}} = \sigma_{\text{drop}} + |f_\theta - \langle f_{\tilde\theta}\rangle|$. Fig. 5 reveals that $\sigma_{\text{drop}}$ contributes to more than $80\%$ of $\sigma_{\text{total}}$ for the three presented datasets and for all applied data splits. A highly similar behavior can be observed for all other

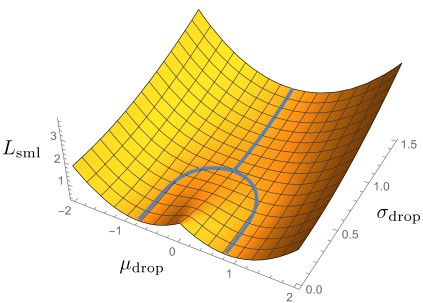

Figure 4: Shown is the value of the loss component $L_2$ as given by eq. (3) over $\mu_{\mathrm{drop}}$ and $\sigma_{\mathrm{drop}}$ describing the implicit dropout ensemble. The blue line shows the position of the minima of $L_2$ for fixed values of $\sigma_{\mathrm{drop}}$. Clearly visible are the global minima at $\sigma_{\mathrm{drop}} = 0$ and the bifurcation at $\sigma_{\mathrm{drop}} = 2/\pi$.

datasets. The analytical consideration in appendix A.1 suggests that for cases without data-inherent uncertainty the SML provides no incentive for $|f_\theta - \langle f_{\tilde\theta}\rangle| > 0$. The same holds true in the presence of aleatoric uncertainty as long as $\sigma_{\mathrm{drop}}$ is comparably large. For aleatoric uncertainty and small $\sigma_{\mathrm{drop}}$ larger $|f_\theta - \langle f_{\tilde\theta}\rangle|$ are favorable. However, as our loss is radial symmetric, all directions are equivalent and initialization and randomness determine the direction of the spread $|f_\theta - \langle f_{\tilde\theta}\rangle|$ for each individual sub-network. This symmetry leads again to a small averaged $|f_\theta - \langle f_{\tilde\theta}\rangle|$. $\sigma_{\mathrm{drop}}$ on the contrary describes the width of a bi-modal set of sub-networks in these cases.

### A.3 DETAILED ANALYSIS OF THE TWO LOSS COMPONENTS

A deeper look into the structure of the second-moment loss is possible if we investigate its behaviour component-wise. To clarify the results presented in Fig. 6, we recall the loss structure as

$$L = L_1 + L_2 = \sum_{i=1}^{M} \left[ a_i^2 + \beta \left( |b_i| - |a_i| \right)^2 \right] \tag{4}$$

with $a_i = f_{\boldsymbol\theta}(x_i) - y_i$ and $b_i = f_{\tilde\theta}(x_i) - f_\theta(x_i)$. Histograms of the $a_i$ (Fig. 6, first column) enable a detailed view on network performance. The uncertainty quality of the networks can be judged by studying the $L_2$ loss term more closely, namely by visualizing histograms of $|b_i| - |a_i|$ (fourth column). The second and third column zoom into $L_2$ and show histograms of the $b_i$ and scatter plots of $(b_i, a_i)$, respectively. Only test datasets are visualized and as we applied $90:10$ train-test splits, this explains the low resolution of some histograms in the first column. All quantities involving $b_i$ require the sampling of sub-networks. We draw 200 sub-networks. This sampling procedure explains the higher plot resolutions in columns two to four.

Qualitatively, we observe that both the $a_i$'s and $b_i$'s are centered around zero which hints at successful optimization of regression performance and of uncertainty quality. Details on how the optimization is technically realistic, can be gained from the scatter plots. They show three qualitative shapes: a 'cross' (first row), a 'line' (second row) and a 'blob' (third and fourth row). The 'cross' occurs for toy-hf and reflects the bi-modal sub-network structure we found in Fig. 1. For an in-detail discussion of the uni- and bi-modality of the second-moment loss landscape see A.1. A 'line' shape reflects that all sub-networks occupy the same minimum given a bi-modal case. Following appendix A.1, a 'blob' indicates a uni-modal case that might be evoked by large standard deviations $\sigma_{\mathrm{drop}}$.

## B EXTENSION TO THE EMPIRICAL STUDY

Accompanying to the evaluation sketched in the body of the paper, section 4, we provide more details on the setup, used benchmarks and measures in the following sub-section. Further information on the toy dataset experiments can be found in section B.2. The same holds for the UCI experiments in section B.3, which we extend by the measures skipped in the main text, and include a description on the used label splits. A close look at the predicted UCI uncertainties (per method) is given via

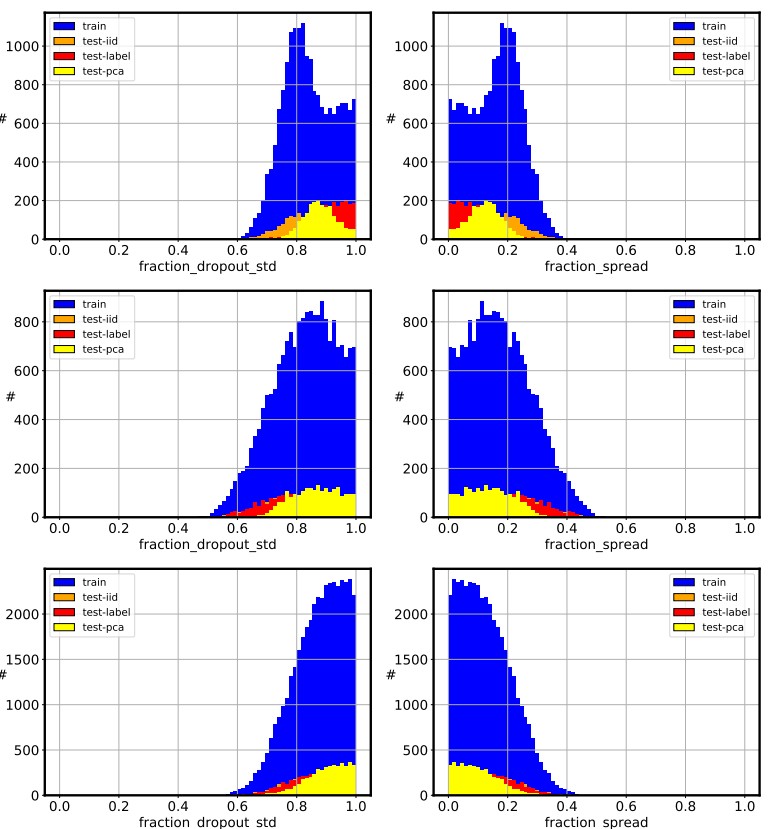

Figure 5: The second-moment loss induces uncertainties $\sigma_{\text{total}} = \sigma_{\text{drop}} + |f_\theta - \langle f_{\hat{\theta}} \rangle|$. The relative contribution of both components ("fraction_dropout_std", "fraction_spread") is shown for three exemplary datasets (top: toy-noise, middle: superconduct, bottom: protein) and i.i.d. (train: blue, test: orange) as well as non-i.i.d. data splits (test-label: red, test-pca: yellow).

scatter plots in section B.4. Details on the SML version of the SqueezedDet are found in the last sub-section.

## B.1 EXPERIMENTAL SETUP

The experimental setup used for the toy data and UCI experiments is presented in three parts: the benchmark approaches we compare with, the evaluation measures we apply to quantify uncertainty, and a description of the neural networks and training procedures we employ.

**Benchmark approaches** We compare dropout networks trained with the SML to archetypes of uncertainty modelling, namely approximate Bayesian techniques, parametric uncertainty, and ensembling approaches. From the first group, we pick MC dropout (abbreviated as **MC**) and its variant last-layer MC dropout (**MC-LL**). While these dropout approaches integrate uncertainty estimation into the very structure of the network, *parametric* approaches model the variance directly as the output of the neural network (Nix & Weigend, 1994). Such networks typically output mean and variance of a Gaussian distribution $(\mu, \sigma)$ and are trained by likelihood maximization. This approach is denoted as **PU** for parametric uncertainty. Ensembles of PU-networks (Lakshminarayanan et al., 2017), referred to as deep ensembles, pose a widely used state-of-the-art method for uncertainty estimation (Snoek et al., 2019). (Kendall & Gal, 2017) consider drawing multiple dropout samples from a parametric uncertainty model and aggregating multiple predictions for $\mu$ and $\sigma$. We denote this approach **PU-MC**. Moreover, we consider ensembles of non-parametric standard networks. We refer to the latter ones as **DEs** while we call those using PU **PU-DEs**. All considered types of net-

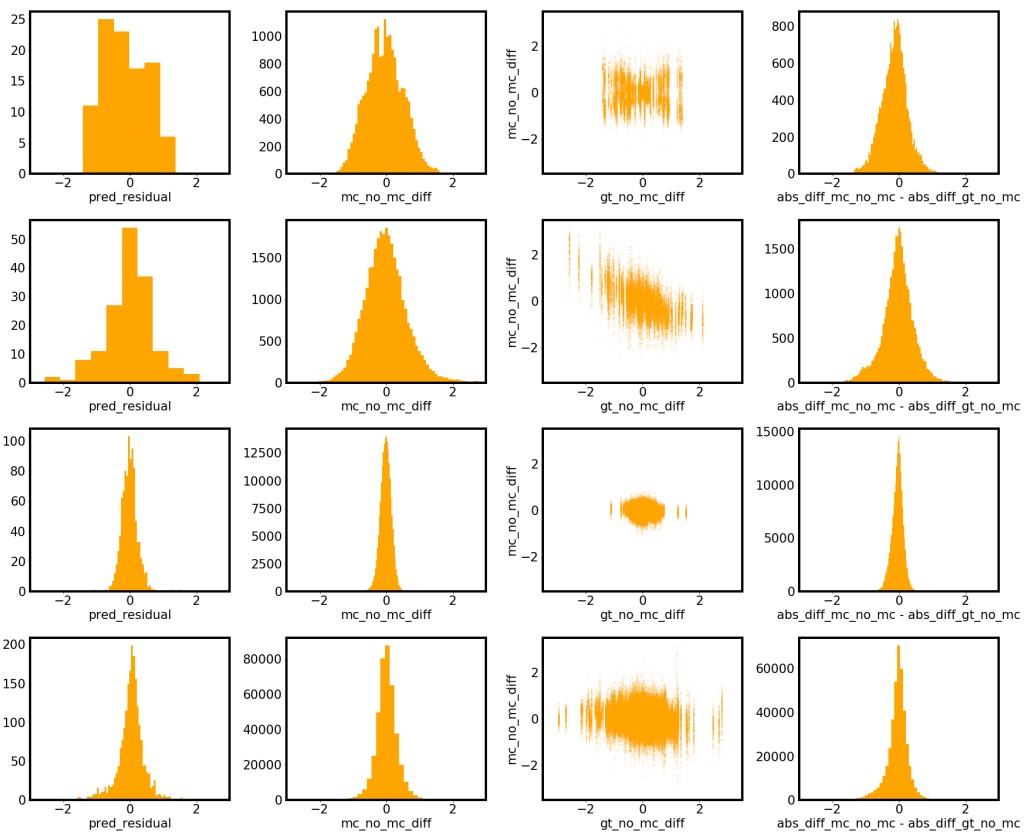

Figure 6: Visualisation of the components (columns) of the second-moment loss for selected test datasets (rows). The prediction residual $f_{\boldsymbol{\theta}}(x_i) - y_i$ (first column), model spread $f_{\tilde{\theta}}(x_i) - f_{\theta}(x_i)$ (second column), a scatter plot of both quantities (third column) and $|f_{\tilde{\theta}}(x_i) - f_{\theta}(x_i)| - |f_{\boldsymbol{\theta}}(x_i) - y_i|$ (fourth column) are shown. The chosen datasets from top to bottom are: toy-hf, wine-red, power, california.

works provide estimates $(\mu_i, \sigma_i)$ where $\sigma_i$ is obtained either analytically (PU), by sampling (MC, MC-LL, SML) or as an ensemble aggregate (DE, PU-DE).

**Evaluation measures** In all experiments we evaluate both regression performance and uncertainty quality. Regression performance is quantified by the root-mean-square error (**RMSE**), $\sqrt{1/N \sum_i (\mu_i - y_i)^2}$ (Bishop, 2006). Another established metric in the uncertainty community is the (Gaussian) negative log-likelihood (**NLL**), $1/N \sum_i \left( \log \sigma_i + (\mu_i - y_i)^2/(2\sigma_i^2) + c \right)$, a hybrid between performance and uncertainty measure (Gneiting & Raftery, 2007), see appendix D.2 for a discussion.[6] The expected calibration error (**ECE**, Kuleshov et al. (2018)) in contrast is not biased towards well-performing models and in that sense a pure uncertainty measure. It reads ECE $= \sum_{j=1}^{B} |\tilde{p}_j - 1/B|$ for B equally spaced bins in quantile space and $\tilde{p}_j = |\{r_i | q_j \leq \tilde{q}(r_i) < q_{j+1}\}|/N$ the empirical frequency of data points falling into such a bin. The normalized prediction residuals $r_i$ are defined as $r_i = (\mu_i - y_i)/\sigma_i$. Further, $\tilde{q}$ is the cdf of the standard normal $\mathcal{N}(0, 1)$ and $[q_j, q_{j+1})$ are equally spaced intervals on $[0, 1]$, i.e. $q_j = (j - 1)/B$. Additionally, we propose to consider the *Wasserstein distance of normalized prediction residuals* (**WS**). The Wasserstein distance (Villani, 2008), also known as earth mover's distance (Rubner et al., 1998), is a transport-based measure denoted by $(d_{\text{WS}})$ between two probability densities, with Wasserstein GANs (Arjovsky et al., 2017) as its most prominent application in ML. For ideally calibrated uncertainties, we expect $y_i \sim \mathcal{N}(\mu_i, \sigma_i)$ and therefore $r_i \sim \mathcal{N}(0, 1)$. Thus we use $d_{\text{WS}}(\{r_i\}_i, \mathcal{N}(0, 1))$ to measure deviations from this ideal behavior. As ECE, this is a pure uncertainty measure. However, it does not use binning and can

---

[6]Throughout the paper, we ignore the constant $c = \log \sqrt{2\pi}$ of the NLL.

therefore resolves deviations on all scales. For example, two strongly ill-calibrated uncertainties ($r_1, r_2 \gg 1$, $r_1 < r_2$) would result in (almost) identical ECE values while WS would resolve this difference in magnitude.

**Technical details**   All investigated neural networks have the same architecture, 2 hidden layers of width 50, and ReLu activations (Glorot et al., 2011). For all dropout-based methods (MC, MC-LL, SML) we set the drop rate to $p = 0.1$. Like MC, SML-trained networks apply Bernoulli dropout to all hidden activations. In the case of MC-LL the dropout is only applied to the last hidden layer. For ensemble methods (DE, DE-PU) we employ 5 networks. For PE networks, we normalize the $\sigma$ value using softplus (Glorot et al., 2011) and optimzie the NLL instead of the MSE. For the optimization of all NNs we use the ADAM-optimizer (Kingma & Ba, 2014) with a learning rate of 0.001. For 'california', the learning rate is reduced to 0.0001 as training of PU and PU-DE is unstable using the standard setup. Additionally, we apply standard normalization to the input and output features of all datasets to enable better comparability.

Number of epochs trained and amount of cross validation differs by the training-set size. We categorize the Toy and UCI datasets as follows: small datasets {toy-hf, yacht, diabetes, boston, energy, concrete, wine-red }, large datasets {toy-noise, abalone, power, naval, california, superconduct, protein } and very large datasets {year }. For small datasets, NNs are trained for $1,000$ epochs using mini-batches of size 100. All results are 10-fold cross validated. For large datasets, we train for 150 epochs and apply 5-fold cross validation. We keep this large-dataset setting for the very large 'year' dataset but increase mini-batch size to 500.

All experiments are conducted on `Core Intel(R) Xeon(R) Gold 6126` CPUs and `NVidia Tesla V100` GPUs. Conducting the described experiments with cross validation on one CPU takes $6\,h$ for toy data, $80\,h$ for UCI regression. And $8\,h$ for object regression on the GPU.

For SML it turns out that as long as $0 < \beta < 1$, the actual value of $\beta$ has only a limited influence on the optimization result, see appendix C for details. Larger $\beta$-values can however favour uncertainty optimization at an expanse of task performance. Throughout the body of the paper we use a conservative value of $\beta = 0.5$.

## B.2   Toy datasets: systematic evaluation

The toy-noise and toy-hf datasets are sampled from $f_{\text{noise}}(x) \sim \mathcal{N}(0, \exp(-0.02\,x^2))$ for $x \in [-15, 15]$ and $f_{\text{hf}}(x) = 0.25\,x^2 - 0.01\,x^3 + 40\,\exp(-(x+1)^2 / 200)\,\sin(3\,x)$ for $x \in [-15, 20]$, respectively. Standard normalization is applied to input and output values. The aggregated measures of the seperate uncertainty methods achieved on these datasets are given in table 3.

Table 3: Regression performance and uncertainty quality of networks with different uncertainty mechanisms. All scores are calculated on the test set of toy-hf and toy-noise, respectively.

| measure | dataset | MC | MC-LL | Ours | PU | PU-DE | DE |
|---|---|---|---|---|---|---|---|
| RMSE ($\downarrow$) | toy-hf | 0.69 | 0.69 | 0.69 | 0.71 | 0.70 | 0.66 |
| NLL ($\downarrow$) | toy-hf | 51.22 | 65.34 | $-0.06$ | $-0.08$ | $-0.08$ | 46.31 |
| ECE ($\downarrow$) | toy-hf | 1.48 | 1.56 | 0.50 | 0.60 | 0.61 | 1.49 |
| WS ($\downarrow$) | toy-hf | 6.13 | 8.17 | 0.26 | 0.26 | 0.26 | 6.14 |
| RMSE ($\downarrow$) | toy-noise | 1.00 | 1.00 | 1.01 | 1.00 | 1.00 | 1.00 |
| NLL ($\downarrow$) | toy-noise | $9.9 \times 10^8$ | $2.3 \times 10^{10}$ | $-0.23$ | $-0.39$ | $-0.39$ | 7942.57 |
| ECE ($\downarrow$) | toy-noise | 1.76 | 1.75 | 0.19 | 0.10 | 0.09 | 1.65 |
| WS ($\downarrow$) | toy-noise | $2.6 \times 10^9$ | $1.7 \times 10^9$ | 0.16 | 0.04 | 0.04 | 72.85 |

## B.3   UCI datasets: RMSEs, NLLs and systematic evaluation

This sub-section provides further details on our UCI experiments covering: an overview on the datasets and splits used for the data-shift studies, further uncertainty measure evaluations (RMSE, NLL, WS), and close with a discussion on the weaker SML results.

**Datasets and data splits** For the UCI regression data, Table 4 provides details on dataset references, preprocessing and basic statistics. Extrapolation and interpolation data-shifts are, technically, introduced by applying non-i.i.d. (independent and identically distributed) data splits. Natural candidates for such non-i.i.d. splits are splits along the main directions of data in input and output space, respectively. Here, we consider 1D regression tasks. Therefore, output-based splits are simply done on a scalar label variable (see Fig. 7, right). We call such a split *label-based* (for a comparable split, see, e.g., Foong et al. (2019)). In input space, the first component of a principal component analysis (PCA) provides a natural direction (see Fig. 7, left). The actual *PCA-split* is then based on projections of the data points onto this first PCA-component.[7] Splitting data along such an direction in input or output space in e.g. 10 equally large chunks, creates 2 *outer* data chunks and 8 *inner* data chunks. Training a model on 9 of these chunks such that the remaining chunk for evaluation is an inner chunk is called data *interpolation*. If the remaining test chunk is an outer chunk, it is data *extrapolation*. We introduce this distinction as extrapolation is expected to be considerably more difficult than 'bridging' between feature combinations that were seen during training.

Table 4: Details on UCI regression datasets. Ground truth (gt) is partially pre-processed to match the 1D regression setup.

| dataset | # features | # datapoints | reference | remarks |
|---|---|---|---|---|
| yacht | 6 | 308 | UCI | |
| diabetes | 7 | 442 | sklearn | |
| boston | 13 | 506 | sklearn | |
| energy | 8 | 768 | UCI | two gt labels: only "cooling load" gt is used |
| concrete | 8 | 1030 | UCI | |
| wine-red | 11 | 1599 | UCI | |
| abalone | 7 | 4176 | UCI | 1st feature is ignored, also called "kin8nm" |
| power | 4 | 9568 | UCI | |
| naval | 16 | 11934 | UCI | two gt labels: only "turbine" gt is used |
| california | 8 | 20640 | sklearn | |
| superconduct | 81 | 21263 | UCI | |
| protein | 9 | 45730 | UCI | |
| year | 90 | 515345 | UCI | |

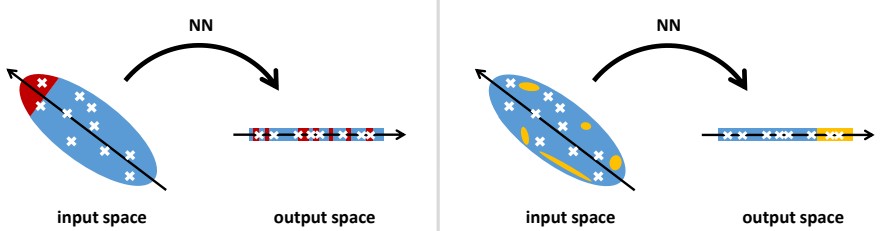

Figure 7: Scheme of two non-i.i.d. splits: a PCA-based split in input space (left) and label-based split in output space (right). While datasets appear to be convex here, they are (most likely) not in reality.

**Regression quality** First, we consider regression performance (see the first and second panel of Fig. 8). Averaging the RMSE values over the considered 13 datasets ('mean' column) yields almost identical results for all uncertainty methods. The only exceptions pose PU, PU-MC and PU-DE with

---

[7]Note that these projections are only considered for data splitting, they are not used for model training.

larger train data RMSEs which could be due to NLL optimization favoring to adapt variance rather than mean. However, this regularizing NLL-training comes along with a smaller generalization gap, leading to competitive test RMSEs. Next, we investigate model performance under data shift, visualized in the third to sixth panel of Fig. 8. Again, regression quality is comparable between all methods. As expected, performances under data shift are worse compared to those on i.i.d. test sets.

**Negative log-likelihoods** For NLL, results are less balanced compared to RMSE (see Fig. 9). PU-DE, PU-MC and the SML-trained network reach comparably small average values on the test set, followed by MC and DE. The average NLL values of MC-LL and PU are above the upper plot limit indicating a rather weak stability of these methods. On PCA-interpolate and PCA-extrapolate test sets, again PU-DE, PU-MC and SML-trained networks perform best. On label-interpolate and label-extrapolate test sets, however, SML-trained networks take the first place with a large margin. The mean NLL values of most other approaches are above the upper plot limit. Note that median results are not as widely spread and PU-DE and SML perform comparably well. These qualitative differences between mean and median behavior indicate that most methods perform poorly 'once in a while'. A noteworthy observation as *stability across a variety of data shifts and datasets* can be seen as a crucial requirement for an uncertainty method. SML-based models yield the highest stability in that sense w.r.t. NLL.

**Wasserstein distances** Studying Wasserstein distances, we again observe equally strong results for PU-DE, PU-MC and SML on train and test data (see first two columns of Fig. 10). PU in contrast possesses a large generalization gap thus yielding weak test set performances. MC, MC-LL, and DE behave consistently weak on train and test sets with MC-LL even falling out of plot range. Under data shift (bottom panel of Fig. 10), the picture remains similar. PU-DE, PU-MC and SML are in the lead and comparably strong for pca-based data shift. On label-based data shifts SML outperforms all other methods by a significant margin. As for NLL, we find these mean values of PU-DE and PU-MC to be significantly above the respective median values indicating again weaknesses in the stability of parametric ensembles.

**Slight overestimation of small uncertainties for SML** The second-moment loss yields weak results on 'yacht', 'energy' and 'naval', the three easiest datasets judging by test set RMSE, compare Table 5. On these datasets neither aleatoric uncertainty nor modelling residuals play a mayor role. In such cases, the second-moment loss seems to cause slightly overshooting uncertainty estimates (compare edges of Fig. 1 for a visual clue), likely due to its sub-network 'repulsion'. Backpropagating not only through $f_{\tilde{\theta}}$ but also through the full network $f_\theta$ in $\mathrm{L}_{sml}$ might mitigate this effect. In practice, slight overestimation of small uncertainties might be acceptable. In contrast, our method performs consistently strong on all more challenging datasets ('california', 'superconduct', 'protein', 'year'). A beneficial characteristic for virtually any real-world task.

### B.4 Residual-uncertainty scatter plots

Visual inspection of uncertainties can be helpful to understand their qualitative behaviour. We scatter model residuals $\mu_i - y_i$ (respective x-axis in Fig. 12) against model uncertainties $\sigma_i$ (resp. y-axis in Fig. 12). For a *hypothetical ideal* uncertainty mechanism, we expect $(y_i - \mu_i) \sim \mathcal{N}(0, \sigma_i)$, i.e. model residuals following the predictive uncertainty distribution. More concretely, 68.3% of all $(y_i - \mu_i)$ would lie within the respective interval $[-\sigma_i, \sigma_i]$ and 99.7% of all $(y_i - \mu_i)$ within $[-3\,\sigma_i, 3\,\sigma_i]$. Fig. 11 visualizes this hypothetical ideal. Geometrically, the described Gaussian properties imply that 99.7% of all scatter points, e.g., in Fig. 12 should lie above the blue $3\sigma$ lines and 68.3% of them above the yellow $1\sigma$ lines. For 'abalone' test data (third row of Fig. 12), PU and SML qualitatively fulfil this requirement while MC and DE tend to underestimate uncertainties. This finding is in accordance with our systematic evaluation. For toy-noise, abalone and superconduct, we qualitatively find PU, PU-DE and SML-trained networks to provide more realistic uncertainties compared to MC, MC-LL and DE (see Fig. 12). The naval dataset poses an exception in this regard as all uncertainty methods lead to comparably convincing uncertainty estimates. The small test RMSEs of all methods on naval (see appendix 8) indicate relatively small aleatoric uncertainties and model residuals. Epistemic uncertainty might thus be a key driving factor and coherently MC, MC-LL and DE perform well.

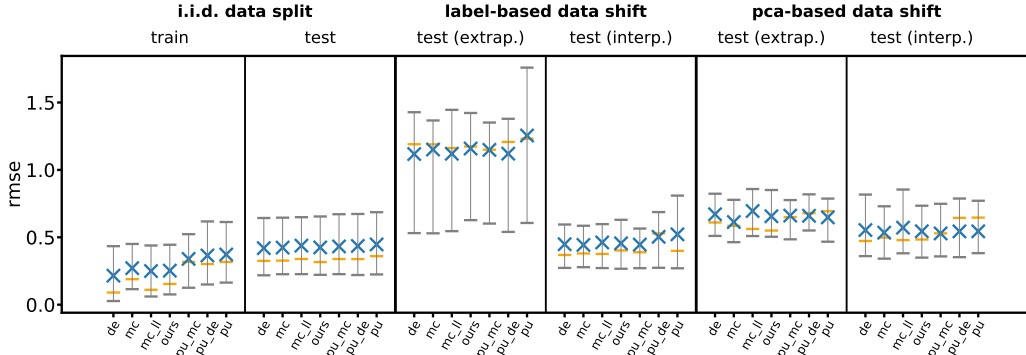

Figure 8: Root-mean-square errors (RMSEs) of different uncertainty methods under i.i.d. conditions (first and second panel) and under various kinds of data shift (third to sixth panel, see text for details). SML ('ours') is compared to 6 benchmark approaches. Each blue cross is the mean over RMSE values from 13 UCI regression datasets. Orange line markers indicate median values. The gray vertical bars reach from the 25% quantile (bottom horizontal line) to the 75% quantile (top horizontal line).

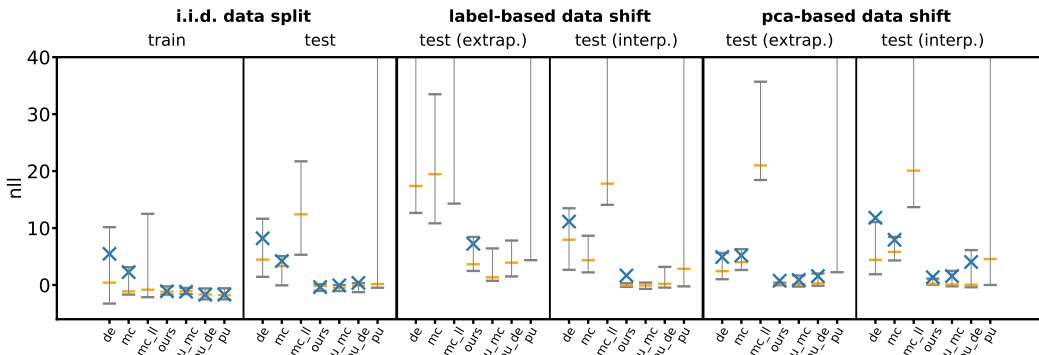

Figure 9: Negative log-likelihoods (NLLs) of different uncertainty methods under i.i.d. conditions (first and second panel) and under various kinds of data shift (third to sixth panel, see text for details). SML ('ours') is compared to 6 benchmark approaches. Each blue cross is the mean over NLL values from 13 UCI regression datasets. Orange line markers indicate median values. The gray vertical bars reach from the 25% quantile (bottom horizontal line) to the 75% quantile (top horizontal line).

Table 5: Regression performance and uncertainty quality of networks with different uncertainty mechanisms. The scores are calculated on the test sets of 13 UCI datasets.

| measure | dataset | DE | MC | MC-LL | Ours | PU-MC | PU-DE | PU |
|---|---|---|---|---|---|---|---|---|
| RMSE (↓) | yacht | 0.04 | 0.07 | 0.06 | 0.08 | 0.08 | 0.05 | 0.06 |
| NLL (↓) | yacht | −3.55 | −2.76 | −2.95 | −2.54 | −2.88 | −3.27 | 0.39 |
| ECE (↓) | yacht | 0.66 | 0.91 | 0.8 | 1.09 | 1.06 | 1.0 | 0.96 |
| WS (↓) | yacht | 0.4 | 0.43 | 0.36 | 0.53 | 0.52 | 0.48 | 1.16 |
| RMSE (↓) | diabetes | 0.92 | 0.84 | 0.94 | 0.87 | 0.79 | 0.81 | 0.82 |
| NLL (↓) | diabetes | 4.5 | 7.34 | 47.46 | 3.19 | 2.43 | 7.55 | 4268.61 |
| ECE (↓) | diabetes | 0.91 | 1.1 | 1.48 | 0.9 | 0.91 | 1.03 | 1.27 |
| WS (↓) | diabetes | 1.6 | 2.47 | 6.9 | 1.49 | 1.29 | 2.09 | 21.42 |
| RMSE (↓) | boston | 0.33 | 0.33 | 0.34 | 0.32 | 0.31 | 0.34 | 0.36 |
| NLL (↓) | boston | 4.47 | 1.89 | 12.41 | −0.08 | 0.27 | 2.88 | 123.59 |
| ECE (↓) | boston | 0.88 | 0.93 | 1.23 | 0.66 | 0.68 | 0.9 | 1.38 |
| WS (↓) | boston | 1.53 | 1.4 | 3.29 | 0.62 | 0.72 | 1.43 | 7.93 |
| RMSE (↓) | energy | 0.08 | 0.08 | 0.08 | 0.08 | 0.09 | 0.1 | 0.12 |
| NLL (↓) | energy | −0.7 | −2.14 | −1.45 | −2.03 | −2.03 | −2.0 | 4.94 |
| ECE (↓) | energy | 0.53 | 0.49 | 0.52 | 0.63 | 0.69 | 0.51 | 1.02 |
| WS (↓) | energy | 0.71 | 0.25 | 0.48 | 0.32 | 0.35 | 0.43 | 2.23 |
| RMSE (↓) | concrete | 0.25 | 0.25 | 0.26 | 0.25 | 0.24 | 0.26 | 0.27 |
| NLL (↓) | concrete | 4.67 | −0.05 | 5.32 | −0.89 | −0.96 | 0.32 | 80.28 |
| ECE (↓) | concrete | 0.66 | 0.52 | 0.73 | 0.39 | 0.4 | 0.54 | 0.79 |
| WS (↓) | concrete | 1.43 | 0.63 | 1.78 | 0.24 | 0.22 | 0.74 | 3.4 |
| RMSE (↓) | wine-red | 0.84 | 0.77 | 0.85 | 0.78 | 0.78 | 0.76 | 0.8 |
| NLL (↓) | wine-red | 1.45 | 3.56 | 20.49 | 0.9 | 6.35 | 4.11 | 1968347.98 |
| ECE (↓) | wine-red | 0.51 | 0.74 | 0.99 | 0.58 | 0.53 | 0.45 | 0.77 |
| WS (↓) | wine-red | 0.73 | 1.43 | 3.74 | 0.58 | 0.78 | 0.82 | 86.81 |
| RMSE (↓) | abalone | 0.64 | 0.65 | 0.65 | 0.65 | 0.64 | 0.64 | 0.64 |
| NLL (↓) | abalone | 29.61 | 19.36 | 67.22 | 0.25 | −0.1 | −0.07 | −0.02 |
| ECE (↓) | abalone | 1.31 | 1.31 | 1.51 | 0.43 | 0.3 | 0.27 | 0.26 |
| WS (↓) | abalone | 4.61 | 4.05 | 7.53 | 0.47 | 0.17 | 0.17 | 0.18 |
| RMSE (↓) | power | 0.22 | 0.23 | 0.23 | 0.22 | 0.23 | 0.22 | 0.22 |
| NLL (↓) | power | 13.17 | 3.32 | 6.48 | −0.84 | −1.0 | −1.0 | −0.98 |
| ECE (↓) | power | 1.06 | 0.93 | 1.03 | 0.21 | 0.23 | 0.15 | 0.12 |
| WS (↓) | power | 2.91 | 1.75 | 2.35 | 0.25 | 0.15 | 0.09 | 0.08 |
| RMSE (↓) | naval | 0.04 | 0.13 | 0.11 | 0.08 | 0.2 | 0.18 | 0.18 |
| NLL (↓) | naval | −2.81 | −1.65 | −0.71 | −1.68 | −1.36 | −1.82 | −2.28 |
| ECE (↓) | naval | 0.97 | 0.54 | 0.66 | 0.9 | 0.64 | 1.03 | 0.75 |
| WS (↓) | naval | 0.55 | 0.49 | 1.04 | 0.5 | 0.4 | 0.53 | 0.46 |
| RMSE (↓) | california | 0.45 | 0.46 | 0.46 | 0.46 | 0.47 | 0.53 | 0.53 |
| NLL (↓) | california | 37.4 | 5.09 | 21.72 | −0.18 | −0.46 | −0.52 | −0.47 |
| ECE (↓) | california | 1.31 | 0.92 | 1.22 | 0.23 | 0.32 | 0.25 | 0.22 |
| WS (↓) | california | 5.07 | 1.85 | 3.94 | 0.27 | 0.2 | 0.15 | 0.15 |
| RMSE (↓) | superconduct | 0.29 | 0.31 | 0.3 | 0.3 | 0.34 | 0.32 | 0.34 |
| NLL (↓) | superconduct | 2.43 | 1.67 | 8.44 | −1.02 | −1.06 | −1.24 | −0.56 |
| ECE (↓) | superconduct | 0.54 | 0.61 | 0.78 | 0.26 | 0.26 | 0.21 | 0.17 |
| WS (↓) | superconduct | 1.16 | 1.14 | 2.1 | 0.16 | 0.16 | 0.14 | 0.24 |
| RMSE (↓) | protein | 0.59 | 0.61 | 0.61 | 0.62 | 0.67 | 0.67 | 0.69 |
| NLL (↓) | protein | 4.38 | 3.7 | 13.14 | 0.07 | −0.07 | −0.12 | 0.14 |
| ECE (↓) | protein | 0.68 | 0.8 | 1.0 | 0.25 | 0.31 | 0.3 | 0.2 |
| WS (↓) | protein | 1.45 | 1.56 | 2.92 | 0.21 | 0.18 | 0.19 | 0.15 |
| RMSE (↓) | year | 0.77 | 0.79 | 0.81 | 0.81 | 0.79 | 0.78 | 0.79 |
| NLL (↓) | year | 11.63 | 15.42 | 1240419.1 | 0.15 | 0.01 | −0.02 | 0.17 |
| ECE (↓) | year | 1.0 | 1.26 | 1.42 | 0.25 | 0.28 | 0.26 | 0.25 |
| WS (↓) | year | 2.49 | 3.45 | 852087.22 | 0.24 | 0.18 | 0.16 | 0.19 |

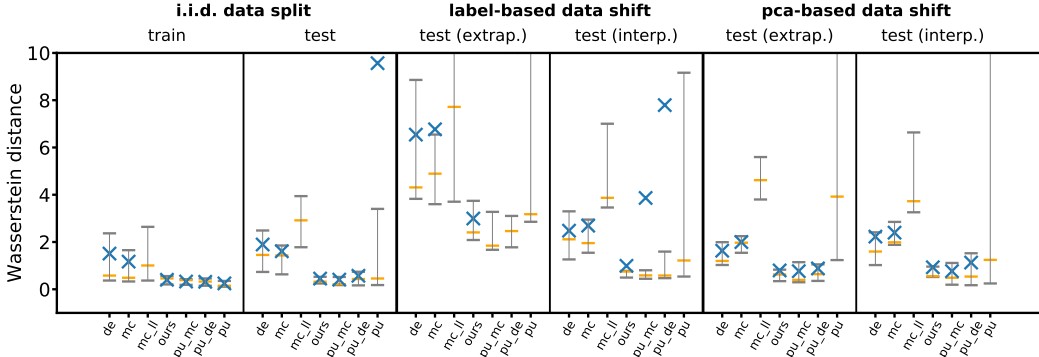

Figure 10: Wasserstein distances of different uncertainty methods under i.i.d. conditions (first and second panel) and under various kinds of data shift (third to sixth panel, see text for details). SML ('ours') is compared to 6 benchmark approaches. Each blue cross is the mean over WS values from 13 UCI regression datasets. Orange line markers indicate median values. The gray vertical bars reach from the 25% quantile (bottom horizontal line) to the 75% quantile (top horizontal line).

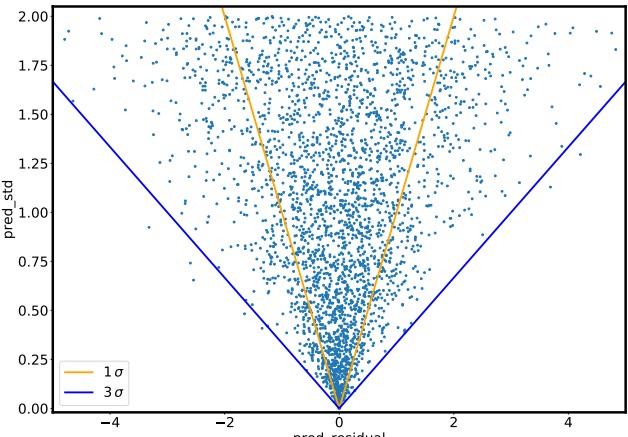

Figure 11: Prediction residuals (x-axis) and predictive uncertainty (y-axis) for a *hypothetical* ideal uncertainty mechanism. The Gaussian errors are matched by Gaussian uncertainty predictions at the exact same scale. $68.3\%$ of all uncertainty estimates (plot points) lie above the orange $1\sigma$-lines and $99.7\%$ of them above the blue $3\sigma$-lines.

The hypothetical ideal residual-uncertainty scatter plot we use in Fig. 11 is generated as follows: We draw 3000 standard deviations $\sigma_i \sim \mathcal{U}(0, 2)$ and sample residuals $r_i$ from the respective normal distributions, $r_i \sim \mathcal{N}(0, \sigma_i)$. The pairs $(r_i, \sigma_i)$ are visualized. By construction, uncertainty estimates now ideally match residuals in a distributional sense. But even in this perfect case, Pearson correlation between uncertainty estimates and absolute residuals is only approximately $55\%$.

### B.5 DETAILS ON SML-SQUEEZEDET

**Architecture** SqueezeDet (Wu et al., 2017) takes an input image and predicts three quantities: (i) $2D$ bounding boxes for detected objects (formalized as a $4D$ regression task), (ii) a confidence score for each predicted bounding box and (iii) the class of each detection. Its architecture is as follows: First, a sequence of convolutional layers extracts features from the input image. Next, dropout with a drop rate of $p = 0.5$ is applied to the final feature representations. Another convolutional layer, the ConvDet layer, finally estimates prediction candidates. In more detail, SqueezeDet

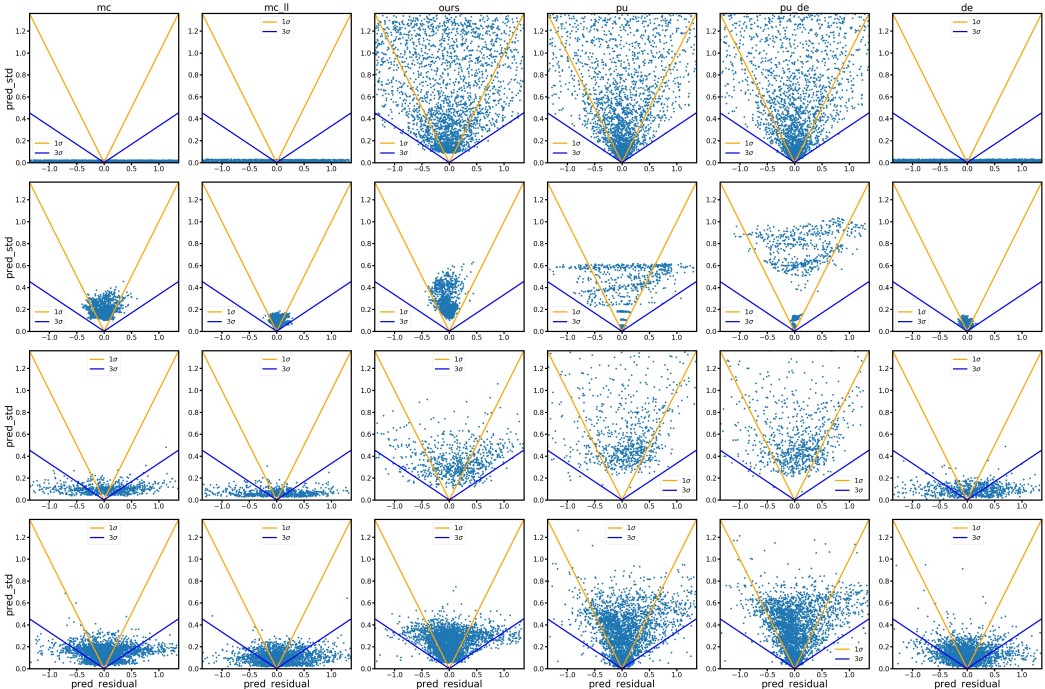

Figure 12: Prediction residuals (respective x-axis) and predictive uncertainty (respective y-axis) for different uncertainty mechanisms (columns) and datasets (rows). Each light blue dot in each plot corresponds to one test data point. Realistic uncertainty estimates should lie mostly above the blue $3\sigma$-lines. The datasets toy-noise, naval, abalone and superconduct are shown, from top to bottom.

predictions are based on so-called anchors, initial bounding boxes with prototypical shapes. The ConvDet layer computes for each such anchor a confidence score, class scores and offsets to the initial position and shape. The final prediction outputs are obtained by applying a non-maximum-suppression (NMS) procedure to the prediction candidates. The original loss of SqueezeDet is the sum of three terms. It reads $L_{\text{SqueezeDet}} = L_{\text{regres}} + L_{\text{conf}} + L_{\text{class}}$ with the bounding box regression loss $L_{\text{regres}}$, a confidence-score loss $L_{\text{conf}}$ and the object-classification loss $L_{\text{class}}$. Our modification of the learning objective is restricted to the L2 regression loss:

$$L_{\text{regres}} = \frac{\lambda_{\text{bbox}}}{N_{\text{obj}}} \sum_{i=1}^{W} \sum_{j=1}^{H} \sum_{k=1}^{K} I_{ijk}[(\widetilde{\delta x}_{ijk} - \delta x_{ijk}^G)^2 + (\widetilde{\delta y}_{ijk} - \delta y_{ijk}^G)^2$$
$$+ (\widetilde{\delta w}_{ijk} - \delta w_{ijk}^G)^2 + (\widetilde{\delta h}_{ijk} - \delta h_{ijk}^G)^2] \tag{5}$$

with $(\widetilde{\delta x}_{ijk}, \widetilde{\delta y}_{ijk}, \widetilde{\delta w}_{ijk}, \widetilde{\delta h}_{ijk})$ and $(\delta x_{ijk}^G, \delta y_{ijk}^G, \delta w_{ijk}^G, \delta h_{ijk}^G)$ being estimate and ground truth expressed in coordinates relative to the k-th anchor at grid point $(i, j)$. Tilde denotes that a variable is subject to dropout-induced randomness. See Wu et al. (2017) for descriptions of all other loss parameters. Applying the second-moment loss component-wise to this 4D regression problem yields

$$L_{\text{regres,SML}} = \frac{\lambda_{\text{bbox}}}{N_{\text{obj}}} \sum_{i=1}^{W} \sum_{j=1}^{H} \sum_{k=1}^{K} I_{ijk}$$
$$[ (\delta x_{ijk} - \delta x_{ijk}^G)^2 + (|\widetilde{\delta x}_{ijk} - \delta x_{ijk}| - |\delta x_{ijk} - \delta x_{ijk}^G|)^2$$
$$(\delta y_{ijk} - \delta y_{ijk}^G)^2 + (|\widetilde{\delta y}_{ijk} - \delta y_{ijk}| - |\delta y_{ijk} - \delta y_{ijk}^G|)^2$$
$$(\delta w_{ijk} - \delta w_{ijk}^G)^2 + (|\widetilde{\delta w}_{ijk} - \delta w_{ijk}| - |\delta w_{ijk} - \delta w_{ijk}^G|)^2$$
$$(\delta h_{ijk} - \delta h_{ijk}^G)^2 + (|\widetilde{\delta h}_{ijk} - \delta h_{ijk}| - |\delta h_{ijk} - \delta h_{ijk}^G|)^2 ] \tag{6}$$

with deterministic network outputs $(\delta x_{ijk}, \delta y_{ijk}, \delta w_{ijk}, \delta h_{ijk})$.

**Details on data split and evaluation**    Our evaluation is based on the KITTI dataset (Geiger et al., 2012), which contains image sequences of driving scenes. We split the original KITTI training set into 3682 train and 3799 test images such that all images of a given sequence are either in the train or in test set (see Xiang et al. (2015)). Predicted bounding boxes are matched with ground truth using a minimum-weight maximum bipartite matching. The edge weights of this bipartite graph are given by the intersection over union (IoU) between ground truth and predicted bounding boxes.

Table 6: Regression performance and uncertainty quality of SqueezeDet-type networks on KITTI train/test data. SML-trained SqueezeDet (ours) is compared with the default SqueezeDet that uses one-layer dropout to estimate uncertainties.

| measure | SqueezeDet | SML-SqueezeDet | SqueezeDet | SML-SqueezeDet |
|---|---|---|---|---|
| | | train | | test |
| mIoU ($\uparrow$) | 0.816 | 0.812 | 0.738 | 0.744 |
| RMSE ($\downarrow$) | 6.418 | 6.862 | 18.225 | 17.492 |
| $\text{NLL}_x$ ($\downarrow$) | 20.070 | 3.488 | 95.621 | 14.252 |
| $\text{NLL}_y$ | 6.064 | 1.453 | 15.376 | 3.709 |
| $\text{NLL}_w$ | 33.715 | 6.593 | 219.778 | 39.870 |
| $\text{NLL}_h$ | 22.594 | 4.128 | 64.451 | 13.667 |
| $\text{ECE}_x$ ($\downarrow$) | 0.919 | 0.431 | 1.145 | 0.783 |
| $\text{ECE}_y$ | 0.955 | 0.520 | 1.140 | 0.775 |
| $\text{ECE}_w$ | 1.025 | 0.512 | 1.296 | 0.915 |
| $\text{ECE}_h$ | 1.085 | 0.751 | 1.209 | 0.864 |
| $\text{WS}_x$ ($\downarrow$) | 2.616 | 0.675 | 5.342 | 1.942 |
| $\text{WS}_y$ | 2.201 | 0.822 | 3.555 | 1.373 |
| $\text{WS}_w$ | 2.422 | 0.785 | 5.813 | 1.975 |
| $\text{WS}_h$ | 2.710 | 1.212 | 4.037 | 1.647 |

## C    STABILITY W.R.T. HYPER-PARAMETER $\beta$

Here, we analyze the impact of the SML-parameter $\beta$ on the uncertainty quality of accordingly trained models. For $\beta = 0.1, 0.25, 0.5, 0.75, 0.9$, we observe only relatively small differences in both ECE (see Fig. 13) and Wasserstein distance (see Fig. 14). $\beta = 0.5$ provides (by a small margin) the best average test set performance in both scores. However, the best-performing $\beta$-value for an individual dataset can vary.

Experiments with $\beta \gg 1$ (not shown here) cause non-convergent training in many cases as primarily uncertainty quality is optimized at the expense of task performance. The opposite extreme case is $\beta = 0$, i.e. network optimization without any dropout mechanism. Applying dropout at inference will therefore cause uncontrolled random fluctuations around the network prediction.

## D    IN-DEPTH INVESTIGATION OF UNCERTAINTY MEASURES

### D.1    DEPENDENCIES BETWEEN UNCERTAINTY MEASURES

All uncertainty-related measures (NLL, ECE, Wasserstein distance) relate predicted uncertainties to actually occurring model residuals. Each of them putting emphasize on different aspects of the considered samples: NLL is biased towards well-performing models, ECE measures deviations within quantile ranges, Wasserstein distance resolves distances between normalized residuals. The empirically observed dependencies between these uncertainty measures are visualized in Fig. 15. Additionally to Wasserstein distances, we consider Kolmogorov-Smirnov (KS) distances (Stephens, 1974) on normalized residuals there. It estimates a distance between the sample of normalized residuals and a standard Gaussian. Different from the Wasserstein distance, the KS-distance is

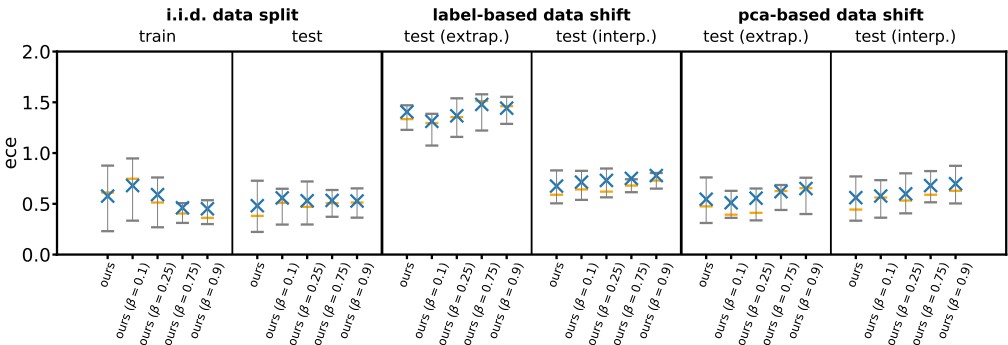

Figure 13: Expected calibration errors (ECEs) for SML-trained networks with hyper-parameters $\beta = 0.1, 0.25, 0.5, 0.75, 0.9$ under i.i.d. conditions (first and second panel) and under various kinds of data shift (third to sixth panel, see text for details). Each blue cross is the mean over ECE values from 13 UCI regression datasets. Orange line markers indicate median values. The gray vertical bars reach from the 25% quantile (bottom horizontal line) to the 75% quantile (top horizontal line).

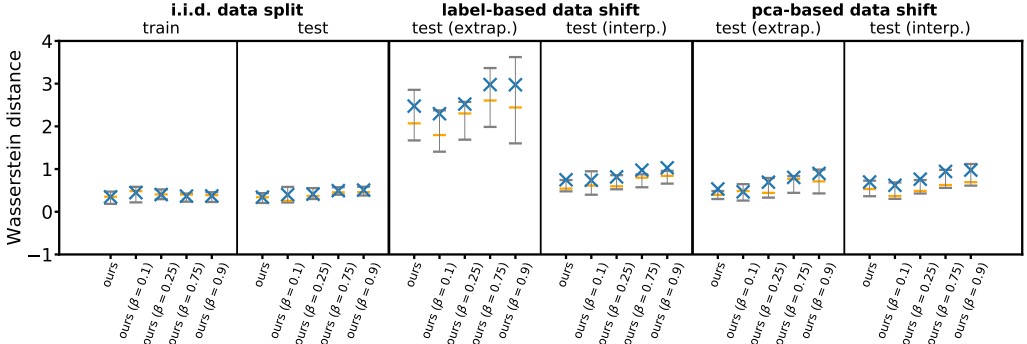

Figure 14: Wasserstein distances for SML-trained networks with hyper-parameters $\beta = 0.1, 0.25, 0.5, 0.75, 0.9$ under i.i.d. conditions (first and second panel) and under various kinds of data shift (third to sixth panel, see text for details). Each blue cross is the mean over WS values from 13 UCI regression datasets. Orange line markers indicate median values. The gray vertical bars reach from the 25% quantile (bottom horizontal line) to the 75% quantile (top horizontal line).

not transport-based but determined by the largest distance between the empirical CDFs of the two samples. It is therefore bounded to $[0, 1]$ and unable to resolve differences between samples that strongly deviate from a standard Gaussian one.

While all these scores are expectably correlated, noteworthy deviations from ideal correlation occur. Therefore, we advocate for uncertainty evaluations based on various measures to avoid overfitting to a specific formalization of uncertainty.

The data splits in Fig. 15 are color-coded as follows: train is green, test is blue, pca-interpolate is green-yellow, pca-extrapolate is orange-yellow, label-interpolate is red and label-extrapolate is light red. The mapping between uncertainty methods and plot markers reads: MC is 'diamond', MC-LL is 'thin diamond', DE is 'cross', PU is 'point', PU-DE is 'pentagon' and second-moment loss is 'square'. The data base of this visualization is toy-noise, toy-hf and 13 UCI regression datasets. Some Wasserstein distances lie above the x-axis cut-off and are thus not visualized.

### D.2 DISCUSSION OF NLL AS A MEASURE OF UNCERTAINTY

Typically, DNNs using uncertainty are often evaluated in terms of their negative log-likelihood (NLL). This property is affected not only by the uncertainty, but also by the DNNs performance. Additionally, it is difficult to interpret, sometimes leading to contraintuitive results, which we want to elaborate on here. As a first example, take the likelihood of two datasets $x_1 = \{0\}$ and $x_2 = \{0.5\}$, each consisting of a single point, with respect to a normal distribution $\mathcal{N}(0, 1)$. Naturally, we find $x_1$ to be located at the maximum of the considered normal distribution and deem it the more likely candidate. But, if we extend these datasets to more than single points, i.e. $\tilde{x}_1 = \{0, 0.1, 0, -0.1, 0\}$ and $\tilde{x}_2 = \{0.5, -0.4, 0, -1.9, -0.7\}$, it becomes obvious that $\tilde{x}_2$ is much more likely to follow the intended Gaussian distribution. Nonetheless, $\mathrm{NLL}(\tilde{x}_2) \approx 1.4 > 0.9 \approx \mathrm{NLL}(\tilde{x}_1)$, where

$$\mathrm{NLL}(y) := \log \sqrt{2\pi\sigma^2} + \frac{1}{N} \sum_{i=1}^{N} \frac{(y_i - \mu)^2}{2\sigma^2} . \tag{7}$$

This may be seen as a direct consequence of the point-wise definition of NLL, which does not consider the distribution of the elements in $\tilde{x}_i$. From this observation also follows that a model with high prediction accuracy will have a lower NLL score as a worse performing one if uncertainties are predicted in the same way. Independent of whether those reflected the "true" uncertainty in either case. This issue can be further substantiated on a second example. Consider two other datasets $z_1, z_2$ drawn i.i.d. from Gaussian distributions $\mathcal{N}(0, \sigma_i)$ with two differing values $\sigma_1 < \sigma_2$. If we determine the NLL of each with respect to its own distribution the offset term in equation (7) leads to $\mathrm{NLL}(z_2) = \mathrm{NLL}(z_1) + \log(\sigma_2/\sigma_1)$ with $\log(\sigma_2/\sigma_1) > 0$. Although both accurately reflect their own distributions, or uncertainties so to speak, the narrower $z_1$ is more "likely". This offset makes it difficult to assess reported NLL values for systems with heteroscedastic uncertainty. While smaller is typically "better", it is highly data- (and prediction-) dependent which value is good in the sense of a reasonable correlation between performance and uncertainty.

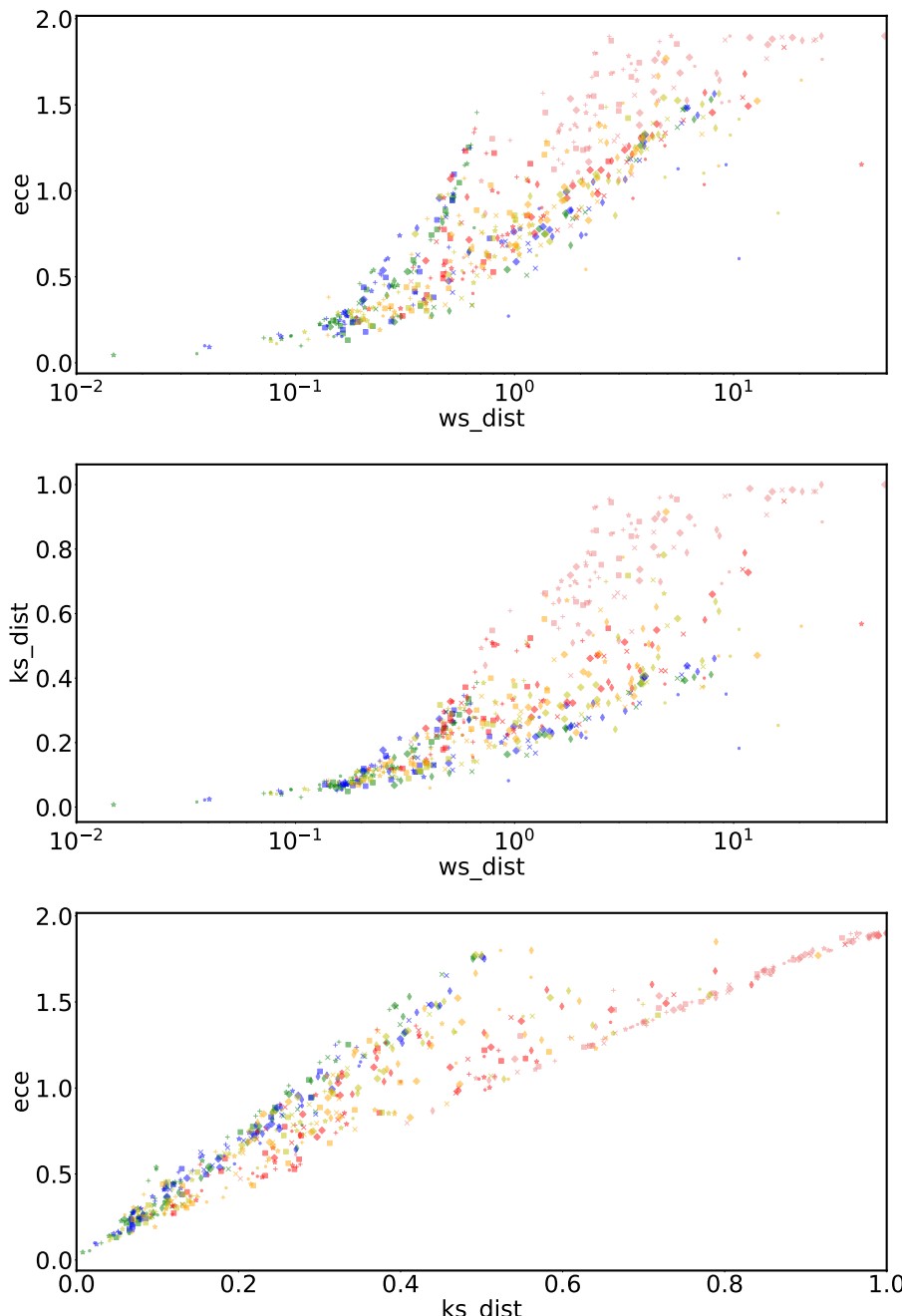

Figure 15: Dependencies between the three uncertainty measures ECE, Wasserstein distance and Kolmogorov-Smirnov distance. Uncertainty methods are encoded via plot markers, data splits via color. Datasets are not encoded and cannot be distinguished (see text for more details). Each plot point corresponds to a cross-validated trained network. The clearly visible deviations from ideal correlations point at the potential of these uncertainty measures to complement one another.

