# OpenReview forum: "Second-Moment Loss: A Novel Regression Objective for Improved Uncertainties"
_ICLR.cc/2021/Conference — Reject_

### Official Review · AnonReviewer4 · 2020-10-18
**A new loss function for better uncertainty estimation.**

**Rating:** 5
**Confidence:** 3

**Review:**

##########################################################################

Summary:

The paper proposes a new objective function to prevent underestimating uncertainties. The authors claimed that the new form leads to state-of-the-art performance in several numerical experiments.

##########################################################################

Pros:

- A new objective function is easy to understand.
- The motivation for the work is sensible.

##########################################################################

Cons:

- Although the motivation of the SML loss in Section 3 is sensible, but the theoretical grounds for the SML loss look somewhat weak. MC dropout (Gal & Ghahramani, 2016) is to maximize the evidence lower bound and learns Bayesian neural networks. What are the properties of the optimal solution of the proposed objective function?
- Following the question, is it restricted to the dropout networks? How can this new loss function be applied other than the dropout networks?
- The authors use the (gaussian) negative log-likelihood with the mean and the variance of the sub-network outputs as one of the evaluation measures. In case of MC dropout, the authors use posterior mean estimates $E(E(Y |\theta, X))$ for $\mu$, but a conditional posterior variance estimates $Var(E( Y | \theta, X))$ for $\sigma^2$. Thus, it makes sense the MC dropout might underestimate uncertainties because $Var(E( Y | \theta, X)) \leq Var( Y | X)$. However, Kendall & Gal (2017), which the author cited in the manuscript, addressed this issue and provided a better uncertainty quantification method. But this method is not considered in the experiments.
- The paper's writing makes it very hard to understand the results. For instance, implementation details for Figure 1 or its pointer are not provided. Also, Figure 3 should be improved. The current presentation is confusing and it's hard to recognize which is better.

##########################################################################

I vote for rejection. I may well have missed some points in my reading, so clarification is welcome.

##########################################################################

After the author response

As the authors address the reviewer's concerns, I changed the rating.

---

> ### Author Response · Authors · 2020-11-18
> **Authors comments in reply to Reviewer4**
>
> We thank the reviewer for the helpful feedback and provide clarifications in the following.
>
> > ###### Cons:
> > ###### Although the motivation of the SML loss in Section 3 is sensible, but the theoretical grounds for the SML loss looks somewhat weak. MC dropout (Gal & Ghahramani, 2016) is to maximize the evidence lower bound and learns Bayesian neural networks. What are the properties of the optimal solution of the proposed objective function?
>
> Studying the loss landscape induced by the second-moment loss is an important topic that we tried to cover in appendix A.1. For 1D data, the bi-modality of  this loss landscape is a key finding (see Fig. 4 and accompanying text for details). We will move parts of this appendix to the main text in the revised version of the paper.
>
> > ###### Following the question, is it restricted to the dropout networks? How can this new loss function be applied other than the dropout networks?
>
> We expect the SML to be “applicable to other models that allow to formulate sub-networks given some kind of mean model“ (see end of section 3). One such model class might be Bayesian approaches beyond MC dropout that learn individual non-binary distributions for each network weight. With slight adjustments our idea of direct variance training carries over to ensemble models without explicit mean model, as well.
>
> > ###### The authors use the (gaussian) negative log-likelihood with the mean and the variance of the sub-network outputs as one of the evaluation measures. In case of MC dropout, the authors use posterior mean estimates $E(E(Y|\theta, X))$ for $\mu$ , but a conditional posterior variance estimates for $\sigma^2$ . Thus, it makes sense the MC dropout might underestimate uncertainties because $Var(E(Y|\theta, X)) \leq Var(Y|X)$ . However, Kendall & Gal (2017), which the author cited in the manuscript, addressed this issue and provided a better uncertainty quantification method. But this method is not considered in the experiments.
>
> It is a valid point that MC Dropout has a tendency to underestimate uncertainties. We believe your argument roughly coincides with our footnote 1 (page 3), which was one of the motivations to implement the SML in the first place. Regarding the mentioned method by Kendall & Gal (2017), it is a combination of MC dropout and parametric uncertainty (PU-MC). We originally decided to benchmark only against “atomistic” methods, as many combinations or variants exist in the literature, see, e.g., [6] for a combination of MC with DE. However, we took up your suggestion and report our findings below (follow link below: Tables 1-3). While PU-MC has comparable performance to the other PU-based methods, and is therefore similar to our SML (follow link below: Tables 1-2), we still observe significant differences due to the different nature of how uncertainty is quantified. This concerns especially the difference for the ETL, where SML still has a significant advantage (follow link below: Table 3).
>
> Additional empirical results: https://i.ibb.co/R2FWb82/Additional-empirical-results-for-the-Second-Moment-Loss.png
>
> [6] Filos, A., Farquhar, S., Gomez, A. N., Rudner, T. G., Kenton, Z., Smith, L., ... & Gal, Y. (2019). A Systematic Comparison of Bayesian Deep Learning Robustness in Diabetic Retinopathy Tasks. arXiv preprint arXiv:1912.10481.
>
> > ###### The paper's writing makes it very hard to understand the results. For instance, implementation details for Figure 1 or its pointer are not provided. Also, Figure 3 should be improved. The current presentation is confusing and it's hard to recognize which is better.
>
> We apologize if the presentation of our work did not fully support an easy understanding of the outlined concepts. We will include a better explanation of Figure 1 in the revised version of the paper. In short, each grey line represents the outputs of one of 200 sub-networks that are obtained by applying dropout-based sampling to the trained full network. For details on the data sets (toy_noise and toy_hf), the neural architecture and the uncertainty methods please refer to section 4 and references therein. Regarding Figure 3, we would like to refer to the second paragraph in section 4.2 that explains the structure of the figure in detail. Moreover, appendix B.3 provides the numerical results underlying it. For the revised version of the paper we are working on a visualization which more strongly emphasizes the SML to allow easier comparison.

---

### Official Review · AnonReviewer2 · 2020-10-29
**Review for Second-Moment Loss**

**Rating:** 4
**Confidence:** 5

**Review:**

The paper proposes an objective function i.e., second-moment loss (SML) to better evaluate the uncertainty based on MC dropout. A full network is used to model the mean, while sub-networks are explicitly used to optimize the model variance.


According to the claim, it seems the main novelty is introducing a new network for variance. First, it is not new that using one network for accuracy (e.g., mean) and another for uncertainty (e.g., variance). There are existing models that should be compared and discussed.

The authors claim that the model can be to adaptive to domain shift, but there is no explanation why the model can do this?

It is unclear or at least not well motivated why the authors proposed the second-term-moment and why it is advantageous than others. The authors should discuss more recent proposed methods in introduction to show the neccessarity or clear advantages for the proposed one.

It is unclear what the gray lines (sub-networks) mean in Fig. 1? And how to obtain the gray lines? It is better to explain in the introduction or caption of fig. 1.

The authors used “prediction uncertainty” and “data-independent uncertainty”. What is the relationship between predictive uncertainty and epistemic uncertainty? What is data-inherent uncertainty? what the relationships between it and aleatoric uncertainty?  Why the authors call the distance $|f_\tilde{\theta} – f_ {\theta} |$ aleatoric uncertainty?

Although the objective is simple, in the current state, the model is still not clear enough to follow.

----------------------------------------------------------------------
Update after rebuttal
----------------------------------------------------------------------
Thanks for the feedback from the authors.

Unfortunately, although some parts are clarified, the main issues still exist. Overall, the novelty is limited and the motivation is still not clear enough.

---

> ### Author Response · Authors · 2020-11-18
> **Authors comments in reply to Reviewer2**
>
> We thank the reviewer for the helpful feedback and provide clarifications in the following.
>
> > ###### The paper proposes an objective function i.e., second-moment loss (SML) to better evaluate the uncertainty based on MC dropout. A full network is used to model the mean, while sub-networks are explicitly used to optimize the model variance.
> > ###### According to the claim, it seems the main novelty is introducing a new network for variance. First, it is not new that using one network for accuracy (e.g., mean) and another for uncertainty (e.g., variance). There are existing models that should be compared and discussed.
>
> We do not make use of an additional network but use sub-networks to encode the variance, similar to MC Dropout. That is, we obtain all measurements from a single DNN via implicit dropout ensembles. Training sampled sub-networks with a separate objective function, that extends the regression one, is to the best of our knowledge a novel approach.
>
> > ###### The authors claim that the model can be to adaptive to domain shit, but there is no explanation why the model can do this?
>
> What we aimed to say is that the model is able to identify out-of-distribution data as it can result from a domain shift. It is typically assumed that dropout-based approaches have a certain “sensitivity” to out-of-distribution data due to a lacking alignment of the sub-networks given unknown input. Our approach models both aleatoric and epistemic uncertainty by means of sub-networks. It is this integration of uncertainty into the very structure of the network that enables the observed good performance under data shift. These rather heuristic arguments aside, we performed out-of-distribution tests on 13 different datasets with 4 different ways to split data (52 tests in total, each cross-validated). Overall, we obtained reliable and competitive results for our method. Moreover, we specifically looked at the safety relevant case of extremely unreliable uncertainties, and could (empirically) show that our method is, by a large margin, more stable than our major competitors (parametric deep ensembles), meaning extremely unreliable uncertainties are more rare.
>
> > ###### It is unclear or at least not well motivated why the authors proposed the second-term-moment and why it is advantageous than others. The authors should discuss more recent proposed methods in introduction to show the neccessarity or clear advantages for the proposed one.
>
> We will extend the motivation of the proposed loss in section 3. For now,  the first paragraphs of section 3 outline the ideas underlying the SML. The paragraphs after equation (1) detail on the SML components and discuss its structure. We would also like to refer to the extended abstract that we uploaded and that further elaborates on the motivation of our approach. The empirical evaluation shows advantages of SML over the most common existing approaches to predictive uncertainty like deep ensembles, MC dropout, and parametric uncertainty. We are happy to include more approaches in our analysis. Which would you consider as the most promising?
>
> > ###### It is unclear what the gray lines (sub-networks) mean in Fig. 1? And how to obtain the gray lines? It is better to explain in the introduction or caption of fig. 1.
>
> We will include a better explanation of Figure 1 in the revised version of the paper. In short, each grey line represents the outputs of one of 200 sub-networks that are obtained by applying dropout-based sampling to the trained full network. For details on the data sets, the neural architecture and the uncertainty methods please refer to section 4 and references therein.

---

> > ### Author Response · Authors · 2020-11-18
> > **Authors comments in reply to Reviewer2 - continued**
> >
> > > ###### The authors used “prediction uncertainty” and “data-independent uncertainty”. What is the relationship between predictive uncertainty and epistemic uncertainty? What is data-inherent uncertainty? what the relationships between it and aleatoric uncertainty? Why the authors call the distance $|f_\theta - f_{\tilde{\theta}}|$ aleatoric uncertainty?
> >
> > The term ‘data-inherent uncertainty’ is used interchangeably with aleatoric uncertainty.* Apologies for not being more clear on this. We will replace the term by the more common expression ‘aleatoric uncertainty’ in the revised version of the paper.
> > Epistemic uncertainty (after training) describes $p(\theta | X)$, i.e. the model parameter uncertainty given the training data. $p(x | \theta, X)$ is marginalized over $p(\theta | X)$ to obtain the posterior predictive distribution $p(x | X)$ (see [5]). Prediction (or predictive) uncertainty describes this resulting distribution. Dropout-based methods approximate this posterior distribution by sampling.
> > We would like to clarify that  $|f_{\theta}(x_i) - f_{\tilde{\theta}}(x_i)|$ is no aleatoric uncertainty. $|f_{\theta}(x_i) - f_{\tilde{\theta}}(x_i)|$ is the distance between the main model’s output without dropout applied and the output of a dropout sub-network. The SML incentivizes this distance to match the prediction residuals $|f_{\theta}(x_i) - y_i|$. This expression, $|f_{\theta}(x_i) - y_i|$, can be seen as an approximate measure of aleatoric uncertainty. The noisy toy dataset (section 4) exemplifies this interpretation: the full network $f_{\theta}$ learns the mean value of the data, approximately $f_{\theta}(x_i) = 0$. Thus $|f_{\theta}(x_i) - y_i| = |y_i|$ depends only on the dataset {$ (x_i, y_i)$} and not on the model $f(x)$ any longer in this case. $|y_i|$ is a proxy to the spread of the dataset at point $x_i$ and thus can be interpreted as an aleatoric uncertainty. We will explain this connection more clearly in the revised version of the paper.
> >
> > *A similar term, ‘data uncertainty’, was used e.g. by the authors of [4].
> >
> > [4] Malinin, A., & Gales, M. (2018). Predictive uncertainty estimation via prior networks. In Advances in Neural Information Processing Systems (pp. 7047-7058).
> >
> > [5] https://en.wikipedia.org/wiki/Posterior_predictive_distribution
> >
> > > ###### Although the objective is simple, in the current state, the model is still not clear enough to follow.
> >
> > We hope that the above replies and the large number of convincing empirical evaluations of the proposed approach summarized in the extended abstract allow for a better and more comprehensive understanding of our approach.

---

### Official Review · AnonReviewer3 · 2020-10-29
**Interesting simple idea, some doubts regarding evaluation**

**Rating:** 6
**Confidence:** 3

**Review:**

##### Summary
The paper proposes a variant of Monte-Carlo Dropout that aims to obtain
better uncertainty estimates by a better adjustment of the output variance. In
concrete, the paper proposes a novel objective function consisting of two
terms. The first term is a simple regression loss between the model expected
output and the data label. The second term, named "second-moment loss" penalizes
the differences in the gap between the expected and sample output and the gap
between the expected output and the data label. In other words, the output
should match the data label in expectation, and the output variance produced by
Dropout "sub-networks" should follow the variance of the prediction residuals,
as a proxy for aleatoric uncertainty.


##### Pros
- I think the idea overall makes sense, it is simple to implement and could be
  an interesting addition to the MC Dropout toolkit.
- Experimental results on toy datasets show the models trained with the proposed
  loss are able to effectively capture the aleatoric uncertainty.
- On real datasets, the proposed method is compared to many uncertainty
  estimation methods and shows competitive performance.

##### Cons
- One potential pitfall could happen when the expected output is far from
  correct due to model limitations and the random "sub-network" predictions
  become heavily biased. I think further discussion on this issue would be
  welcome.
- The paper is in general correctly structured but leaves some important details
  to the appendix. Maybe it is because my lack of familiarity with these
  evaluation metrics, but in my opinion it may be good to clarify them in the
  main paper. For example:
    - The "normalized residuals" $r_i$ in page 6 are not properly introduced in
      the main text.
    - Isn't the ECE, as defined in the appendix, a measure of fit of the
      residuals to a uniform distribution? If yes, why is this a reasonable
      assumption and why would this indicate a correctly characterized
      uncertainty?
    - Are the $\mu_i$ and $\sigma_i$ in the evaluation metrics computed from the
      empirical predictions or from the data? If computed from the predictions,
      then the proposed Wasserstein metric looks like a measure of
      Gaussianity. This would be in contradiction with the ECE metric, and the
      same questions as in the previous point would apply: why is Gaussianity a
      reasonable model for the residuals and a desiderata for the output
      distribution? what is the intuition for why this test indicates good
      uncertainty estimates?
- For a fair comparison with ensemble methods, the total number of parameters
  and operations should be taken into account.


**************
After Rebuttal:
I thank the authors for their extensive answers and clarifications.
Overall, I maintain my positive outlook on this work. Although theoretical justification could be improved, I think the experiments do signal that there is something interesting and valuable in this simple approach for characterizing uncertainty.

---

> ### Author Response · Authors · 2020-11-18
> **Authors comments in reply to Reviewer3**
>
> We thank the reviewer for the helpful feedback and provide clarifications in the following.
>
> > ###### Cons
> > ###### One potential pitfall could happen when the expected output is far from correct due to model limitations and the random "sub-network" predictions become heavily biased. I think further discussion on this issue would be welcome.
>
>
> We asked ourselves a similar question. That is why we considered the high-frequency toy experiment (see section 4.1) where the dataset has a much higher complexity than the deliberately small networks. Our SML and PU are still capable of modelling the occurring network errors. We suspect that taking only the absolute error values gives PU and SML an advantage in modelling them, as those values more easily neglect the highly oscillatory structure of the original data. Thus, even small networks with low expressiveness can provide good uncertainties. We will stress this aspect in the updated version of our paper.
>
> > ###### The paper is in general correctly structured but leaves some important details to the appendix. Maybe it is because my lack of familiarity with these evaluation metrics, but in my opinion it may be good to clarify them in the main paper. For example:
> > ###### The "normalized residuals" in page 6 are not properly introduced in the main text.
>
>
> We apologize for this mistake in the organization of our paper. We will move the definition of the normalized prediction residuals ($ r_i = (\mu_i - y_{gt,i}) / \sigma_i $) from appendix B.1 to the main text.
>
> > ###### Isn't the ECE, as defined in the appendix, a measure of fit of the residuals to a uniform distribution? If yes, why is this a reasonable assumption and why would this indicate a correctly characterized uncertainty?
>
> We will add additional clarification regarding this point to the main text. In short, we assume that prediction residuals are not uniformly but Gaussian distributed (for a discussion of Gaussianity see the comment after the next one). ECE, however, is not directly based on the prediction residuals but on the quantiles of the assumed Gaussian residual distribution. Quantiles of any probability distribution are - by definition - uniformly distributed. Therefore deviations from the uniform distribution in quantiles are a measure of non-Gaussianity of the residuals. For a formal definition of ECE, see e.g. [1].
>
> [1] Guo, C., Pleiss, G., Sun, Y., & Weinberger, K. Q. (2017). On calibration of modern neural networks. arXiv preprint arXiv:1706.04599.
>
> > ###### Are the $\mu_i$ and $\sigma_i$ in the evaluation metrics computed from the empirical predictions or from the data?
>
> They are computed from the empirical predictions. For dropout-based methods, $\mu_i$ and $\sigma_i$ are estimated from a sample of empirical (dropout-based) predictions $\{y_k\}, k=1,..,N$ with N being the number of (dropout) forward passes. For the other methods please see the explanations in appendix B.1. We will add these explanations to the main text in the revised version of our paper.

---

> > ### Author Response · Authors · 2020-11-18
> > **Authors comments in reply to Reviewer3 - continued**
> >
> >
> > > ###### If computed from the predictions, then the proposed Wasserstein metric looks like a measure of Gaussianity. This would be in contradiction with the ECE metric, and the same questions as in the previous point would apply: why is Gaussianity a reasonable model for the residuals and a desiderata for the output distribution? what is the intuition for why this test indicates good uncertainty estimates?
> >
> > In both cases Gaussianity is assumed. Evaluations of uncertainties typically require assumptions on a model distribution. Uncertainty scores quantify network predictions relative to these distributions. Due to its universality, most uncertainty quantifications rely on a Gaussian distribution, see e.g. the (Gaussian) NLL in [2], [3].
> >
> > There are two reasons that favour the Gaussian model assumption: at first if one assumes that deviations have multiple sources/causes one is tempted to consider the resulting distribution as Gaussian (central limit theorem). Secondly, this assumption is linked with the type of loss considered, least square regression as commonly used for regression assumes a Gaussian nature of potential noise to work adequately.
> >
> > While it might be possible to find more appropriate dataset-specific model distributions, this would increase modelling complexity. Candidates for such more complex distributions are heavy-tailed or mixture distributions (see e.g. [3] (p. 3, last sentence)).
> >
> > [2] Gal, Y., & Ghahramani, Z. (2016, June). Dropout as a Bayesian approximation: Representing model uncertainty in deep learning. In International Conference on Machine Learning (pp. 1050-1059).
> >
> > [3] Lakshminarayanan, B., Pritzel, A., & Blundell, C. (2017). Simple and scalable predictive uncertainty estimation using deep ensembles. In Advances in Neural Information Processing Systems (pp. 6402-6413).
> >
> > > ###### For a fair comparison with ensemble methods, the total number of parameters and operations should be taken into account.
> >
> > We employ the same neural architecture (except for minor changes like a second output for PU) for all uncertainty methods. PU-DE ensembles consist of 5 networks. All sampling-based methods use 50 dropout forward passes. Thus, ensembles require more storage, while sampling methods require more computations. While it might be technically possible to equalize storage and computational requirements  for all methods, this causes conceptual problems as e.g. an ensemble of 5 networks has another expressivity than one network that has as many parameters as the entire ensemble. For a short discussion of techniques to reduce the ‘footprint’ of the SML we would like to refer to section 5.

---

### Author Response · Authors · 2020-11-18
**Extended paper abstract**

Following the feedback of the reviewers we provide an “extended abstract” to allow a better overview on the content and achievements of our paper. All references in the text below are made with respect to the submitted paper and its appendix (see the pdf in the supplementary material zip).

**Synopsis / Motivation**

While Deep Neural Networks (DNNs) find application in an ever increasing amount of tasks, ensuring their reliability, especially for safety-critical tasks, is still an open problem. Outputting an additional uncertainty information together with the prediction could provide a form of “self-assessment” to alleviate this. And, a plethora of methods are available to obtain such uncertainty, each with its own drawbacks and strengths. In addition to accurate predictions these methods are also required to provide reliable, i.e. realistic, uncertainties, which can be used to decide whether a given prediction is trustworthy.

We propose a modified learning objective, the second-moment loss (SML), to increase the reliability of MC Dropout-based uncertainty approaches. While MC Dropout is fairly widespread, one typically finds that it does not capture uncertainty stemming from data-inherent uncertainty, e.g. from noisy data, well. For regression tasks, our approach addresses this without  (1) changing the underlying architecture, (2) losing the “dropout nature” of the uncertainty measurement, (3) decreasing the performance and (4) at no additional cost to inference. This is achieved, loosely speaking, by coupling the spread of the dropout-induced distribution to the network residuals (deviations between prediction and ground truth) during training, see equation (1) in the paper. Our extensive evaluation (continue below for detail) shows that we reach in-data uncertainty reliability comparable to deep ensembles, an expensive state-of-the-art method, with additional gains regarding the robustness of our uncertainties when confronted with structurally new, i.e. out-of-distribution, data.

---

> ### Author Response · Authors · 2020-11-18
> **Extended paper abstract - continued**
>
> **Experiments**
>
> We provide three groups of experiments: the majority of evaluations is performed on a selection of UCI datasets, accompanied by didactic toy experiments and a more complex task, 2D bounding box detection on KITTI using SqueezeDet. Our experiments are reproducible with the code provided in the supplementary material zip file. Here, we summarize the results, mainly from the body of the paper.
>
> *Benchmarks and Measures*
>
> We compare our own method (SML) to a variety of differing methods representing various classes of approaches: Bernoulli MC Dropout (MC), its “last-layer” version (MC-LL), parametric uncertainty (PU) directly predicting uncertainties trained via negative log-likelihood, and deep ensembles without (DE) and with included parametric uncertainty prediction (DE-PU).
>
> For measures we focus in the body of the paper on the expected calibration error (ECE) and report the negative log-likelihood (NLL) as well as a novel Wasserstein distance-based measure (WS) in the appendix for all used methods. In terms of performance the root-mean-square error (RMSE) is given there as well. All results are obtained using (at least) 5-fold cross validation.
>
> *Toy Dataset*
>
> A didactical study on uncertainty stemming from noisy or highly oscillatory data. While this kind of noise is not captured by MC, compare Fig. 2, both SML and PU/DE-PU can accurately reflect it. Please recall that PU and SML are entirely different approaches to obtain uncertainty estimates, as the latter is based on MC dropout-like inference. Appendix A features more theoretical investigations of the SML mechanics.
>
> *UCI Datasets*
>
> For a more realistic assessment of our loss, we benchmark its performance on 13 UCI regression datasets. While the in-data tests are performed on i.i.d. test-sets we also perform four types of out-of-distribution (OOD) tests: we split the existing data either by label or according to its major (pca) component and perform on these sets either interpolation by not training on inner segments or extrapolation when neglecting the edges (details and motivation: appendix B.3). The SML consistently provides a performance met only by DE-PU for the ECE, NLL and WS, compare Figs. 3, 9, 10 respectively. Regarding performance (RMSE) most methods are comparable, i.e. our new SML objective does not provide the improved uncertainty at a cost to performance, see Fig. 8. For easier comparison the in-data test results are also reported in Table 5. Further details on this study can be found in appendix B.
>
> We additionally investigate the worst-case uncertainties, i.e. those cases where the mismatch between predicted reliability (uncertainty) and actual prediction offset (to ground truth) were largest. From a safety point of view, such extremal events carry a large weight. Consider for example a self-driving vehicle, while minor deviations in the detections of objects / pedestrians can be compensated by, e.g., a safety margin, recognizing strongly misaligned detections is critical. In this respect we can report significant gains, see Table 1, compared to PU-DE (PU would perform even worse), especially when looking at the OOD tests.
>
> *SqueezeDet*
>
> To demonstrate the feasibility of our SML objective for real world applications, we employ SqueezeDet, a 2D bounding box detector, and train it on KITTI, a real world dataset for autonomous driving. We keep the existing dropout layer of SqueezeDet active during inference to perform MC Dropout forward passes. Details of the implementation can be found in appendix B.5. For the training we either use a “vanilla” loss, to obtain MC Dropout (MC) uncertainties, or the SML modification. While both networks have comparable performance (RMSE, mIoU) uncertainty-related scores (ECE, NLL, WS) improve substantially for the SML, see Tables 2 and 6.

---

### Author Response · Authors · 2020-11-24
**Upload of Revised Paper**

Following the discussions with the reviewers, we revised Figure 3 and also Figures 8-10 (appendix) to provide an improved visualization of our main results. Moreover, we included the suggested method by Kendall & Gal (2017) as a further benchmark.  Several clarifying explanations were added to the paper. For convenience, we included the appendix in the main pdf.

---

### Decision · Program_Chairs · 2021-01-07
**Final Decision**

**Decision:**

Reject

**Comment:**

Dear Authors,

Thank you very much for your very detailed feedback and also updating the manuscript in the rebuttal phase. Your effort has highly contributed to clarifying some of the concerns raised by the reviewers and improving our understanding of your work.

On the other hand, we still think that the current work has rather limited novelty, and motivation and theoretical justification need to be further enhanced to be accepted for ICLR.

For these reasons, I suggest rejection of this paper, in comparison with many other strong submissions. The reviewers added further comments after receiving your feedback. I hope their comments are useful for improving your work for future publication.